# Open-Set Domain Adaptation Under Background Distribution Shift: Challenges and A Provably Efficient Solution

## Abstract

In Open-Set Domain Adaptation (OSDA) we wish to perform classification in a target domain which contains a novel class along with $k$ non-novel classes. This work formally studies OSDA under the assumption that classes are separable, and the supports of source and target domains coincide, while other aspects of the distribution may change. We term such a distribution shift as background shift. We develop a simple and scalable OSDA method that attains robustness to background shift and is guaranteed to solve the problem, while showing that it cannot be solved under weaker conditions for OSDA studied in the past, particularly in the presence of covariate shift. We formally define the realistic assumptions of background shift within the scope of OSDA problem that the previous literature has either overlooked or not explicitly addressed. In a thorough empirical evaluation on both image and text data, we observe that existing OSDA methods are not robust to the distribution shifts we consider. Our proposed solution jointly learns representations via concurrently learning to classify known categories and detect novel ones using methods with formal guarantees. The results demonstrate that optimizing these two objectives in unison leads to mutual performance improvements contrary to what might be expected when objectives are considered independently. Our rigorous empirical study also examines how OSDA performance under distribution shift is affected by parameters of the problem such as the novel class size. Taken together, our observations emphasize the importance of formalizing assumptions under which OSDA methods operate and to develop appropriate methodology that is capable of scaling with large datasets and models for different scenarios of OSDA.

## 1 Introduction

Adapting Machine Learning models to shifts in the data distribution is pivotal to ensuring their robustness and safety in real world applications (Quinonero-Candela et al., 2008; Koh et al., 2021) for fields like healthcare (Finlayson et al., 2021), autonomous driving (Filos et al., 2020; Wong et al., 2020) and more broadly in computer vision (Bendale & Boult, 2016). Maintaining robustness to distribution shift in classification problems includes both identifying familiar objects under new conditions, while detecting the long tail of object categories that has not been observed in the past. In this work we study the problem of adaptation when these two types of distribution shift occur concurrently. Namely, the appearance of novel categories, known as Open-Set Domain Adaptation (OSDA) (Panareda Busto & Gall, 2017), and shifting distributions of known classes.

Let our training data be $S_\mathcal{S}$. The core task here is to detect the novel classes in target data $S_\mathcal{T}$ collected under some conditions different from those observed at training time (Panareda Busto & Gall, 2017) while maintaining good performance over the existing classes. The emergence of a novel class in itself constitutes a shift between $S_\mathcal{S}$ and $S_\mathcal{T}$, however, in real world scenarios *it is likely not the only distribution shift that occurs* (Garg et al., 2022; Wald & Saria, 2023). Our study concentrates on scenarios where the novel class exhibit a notable degree of separability from the other known classes, and a distribution shift with overlapping supports between source and target distributions exists within the known classes, which we refer to as 'background shift'. This is arguably the most common case in practice like medical applications, e.g. classifying histopathology slides for known and novel tumor cells.

It is crucial to highlight the understudied aspects of OSDA in the existing literature as a motivation:

The scenario of concurrent shift in known classes, and the appearance of novel ones, is challenging as they require making clear assumptions on the nature of shift or novelty (David et al., 2010; Fang et al., 2022). However, **methods that are explicit about the assumptions under which they are guaranteed to solve these problems, are often not those that obtain competitive performance in practice, especially with large scale data and models.** Therefore in this work we seek to study and improve the applicability of algorithms that have formal guarantees under distribution shift to large scale problems, under realistic conditions.

Most research on OSDA deals with settings where the distribution of known classes does not shift Ge et al. (2017); Neal et al. (2018); Lin & Xu (2019); Chen et al. (2021); Zeng et al. (2021); Vaze et al. (2022); Esmaeilpour et al. (2022) while some methods like Liang et al. (2021); Li et al. (2021) focus on OSDA under a broad umbrella of "domain shifts". Office-Home Venkateswara et al. (2017) is a popular benchmark used to evaluate methods that adapt to a target domain like image sketches ($S_{\mathcal{T}}$) while $S_{\mathcal{S}}$ consists of real images. Hence, **OSDA methods that work in practice often lack clear technical assumptions about the distribution shift leading to inconsistent performances across the varying nature of shifts.** Commonly studied shifts like in Office-Home dataset are qualitatively different from shifts we study here, which include shifts in histopathology slides due to changes in certain patient subpopulation or demographics where the support of the source domain must overlap with that of the target domain, see assumption 1 in Section 3. Although this assumption may restrict the shifts we study to a form of domain shift, our empirical observations reveal that methods effective for domain shift do not necessarily perform well under the specific shifts considered in our assumption 1. Therefore, our study aims to deepen understanding of how such shifts affect our capability to identify novel classes and accurately classify known classes in $S_{\mathcal{T}}$ by addressing correlated tasks while learning a shared representation (Caruana, 1997; Maurer et al., 2016; Baxter, 1997; 2000).

Successful monitoring of novel classes facilitates making critical decisions like updating models to accommodate these new classes, or defer classifications to humans when objects are suspected to be unknown to a classifier to increase their reliability (Amodei et al., 2016; Hendrycks & Gimpel, 2016). **Existing OSDA methods do not sufficiently address the impact of varying proportions of novel samples on OSDA performance.** Hence, we empirically validate our method in such under-explored situations where the novel class is present in very small proportions, which posits greater challenges than dealing with large quantities of unknown classes compared to the size of known classes.

Large foundation models rely on large image-text pretraining datasets, which are often unavailable in specialized fields like medical applications. For example, missing captions of histopathology slides or detailed disease descriptions may restrict training or fine-tuning of such models. Moreover, novel tumors or diseases might not appear in the pretraining data, making validation on natural images benchmark insufficient for open-set problems. This underscores **the need for a fair OSDA benchmark that excludes novel classes from pretraining dataset to validate the legitimacy of large foundation model based solutions for OSDA**. Building such a dataset would require significant efforts and involve challenges beyond the scope of this study.

Our contributions to these aspects of OSDA are thus the following:

- **Mitigate the impact of understudied aspects of OSDA problem:** Existing literature in OSDA under distribution shift within the non-novel instances is either very limited or primarily focuses on specific scenario of label shift Garg et al. (2022). We observe that conditions deemed sufficient for OSDA under label shift may not suffice under other shifts like covariate shift (see Lemma 1). We also determine that the condition of separability (Assumption 1) along with the definition of background shift is applicable widely in real world scenarios. Furthermore, our experiments primarily focus on long-tail setting of novel objects characterized by low proportions of novel class samples.

- **Leverage and improve principled methods for novelty detection:** We improve the existing methods having guarantees for novelty detection to scale effectively with large models and for complex, high-dimensional datasets such as images and text. To mitigate distribution shift within non-novel classes, we further optimize and extend the constrained learning rule proposed in (Wald & Saria, 2023). Finally, we adapt these existing PU learning/novel subpopulation detection methods for the task of OSDA under background shift by learning shared representations.

- **Rigorous analysis across diverse data modalities and datasets:** Following these improvements, we present strong empirical results and rigorous analysis, demonstrating that, when combined with

a joint learning approach, these methods can outperform most existing baselines for OSDA under background shift. Particularly, our enhanced adaptation of the constrained learning rule using shared representations CoLOR shows strong empirical performance across diverse datasets including CIFAR100 Krizhevsky (2009), Amazon Reviews Ni et al. (2019) and SUN397 Xiao et al. (2010). These results demonstrate that our method outperforms all other baseline methods used in this study on most of the OSDA scenarios under background shift.

## 2 RELATED WORK

The OSDA problem lies at the intersection of novelty detection and domain adaptation, while our work also leverages and builds upon Multitask learning.

**Open Set Domain Adaptation:** OSDA has recently garnered significant research attention Wen & Brbic (2024); Qu et al. (2023); Ge et al. (2017); Vaze et al. (2022); Liu et al. (2018); Xu et al. (2019); Saito et al. (2018); Panareda Busto & Gall (2017); Choe et al. (2024); Hur et al. (2023); Zhu et al. (2023). However, none of these studies extend beyond the traditional OSDA setting to address the background shift with the separability assumption. Bendale & Boult (2016) initially proposed Openmax algorithm deriving from Extreme Value Theory followed by data augmentation based G-Openmax method proposed in Ge et al. (2017) which uses generative models to generate unknown samples. There is good line of work based on generative models like Neal et al. (2018), OpenGAN using real open-set images for selecting efficient models and was proposed in Kong & Ramanan (2021). Vaze et al. (2022) leverages well trained closed set classifier for open-set recognition task. Furthermore, there has been OSDA literature that use trained image caption generator such as in Esmaeilpour et al. (2022) which proposes to employ zero-shot classification through pretrained multi-modal representation learning (CLIP). Finally, Qu et al. (2023) propose multi-modal large foundation models (LFMs) like ChatGPT, DALL-E and CLIP for OSDA. However, note that there are several implicit assumptions and limitations that one is subject to when using such LFMs, such as, no access to the pretraining data. This makes it infeasible to curate novel classes and design distribution shift for OSDA benchmark that accommodates such LFM based solutions.

**Multi-Task Learning:** Multi-Task Learning (MTL) harnesses knowledge from related tasks to enhance the generalization of machine learning models Caruana (1997). It involves acquiring shared representations through simultaneous learning from these tasks. Previous research has extensively explored MTL using hard/soft parameter sharing and has provided Bayesian or formal models within the PAC setting Baxter (1997; 2000).

**Out-of-Distribution (OOD) Detection:** Most of the literature on OOD detection deals with the setting where a learner is shown $S_{\mathcal{T}}$ only at test time (Ruff et al., 2021; Esmaeilpour et al., 2022) and a OOD/anomaly score (i.e. a score that ranks examples as more or less likely to be OOD) must be determined prior to observing them. Previous work find significant differences in performance when scores are calculated on different representations (e.g. pre-trained features, self-supervised, or obtained from a classifier/auto-encoder) which play a crucial role here Shen et al. (2022).

**Novel Class Discovery:** Theoretical guarantees for this problem rely on distributional assumptions, and making assumptions is necessary in order to learn a classifier with non-trivial guarantees in this scenario (Bekker et al., 2019; Wald & Saria, 2023). While the majority of methods for this task are heuristic and lack theoretical guarantees, positive and unlabelled (PU) learning based are well studied providing effective solutions to novelty detection (Blanchard et al., 2010; Garg et al., 2021; du Plessis et al., 2014; Elkan & Noto, 2008). Garg et al. (2022) proposed a $k$-way PU learning based solution under a label-shift assumption, i.e. $P_{\mathcal{S}}(\mathbf{x} \mid Y = y) = P_{\mathcal{T},[k]}(\mathbf{x} \mid Y = y)$ for all $y \in [k]$ where $[k]$ is the set of known classes. [1].

**PU-Learning Under Distribution Shift:** Under no distribution shift, solutions based on adjustments to a "Domain Discriminator" (i.e. models that are trained to distinguish $P_{\mathcal{S}}$ and $P_{\mathcal{T}}$) are rather effective (Garg et al., 2021; du Plessis et al., 2014; Elkan & Noto, 2008), however they can underperform when $P_{\mathcal{S}} \neq P_{\mathcal{T},[k]}$ even when assumptions such as separability (see 1) holds. Under separability the problem can be solved given infinite data Gerych et al. (2022), and Wald & Saria (2023) give an algorithm with finite-sample guarantees that we will extend here to the case of OSDA under background shift.

---

[1]They rely on some additional distributional assumptions, which we discuss and compare to our in 3.1

## 3 OPEN-SET DOMAIN ADAPTATION UNDER BACKGROUND SHIFT

For a prediction task with $k$ classes, we are interested in detecting the emergence of a novel class where $Y = k + 1$, while classifying a set of known classes $Y = 1, \ldots, k$. To treat this formally, we assume a learner is provided with two datasets, the training set $S_{\mathcal{S}} = \{\mathbf{x}_i, y_i\}_{i=1}^{n_{\mathcal{S}}}$ and an unlabelled target dataset $S_{\mathcal{T}} = \{\mathbf{x}_i\}_{i=1}^{n_{\mathcal{T}}}$. The datasets $S_{\mathcal{S}}, S_{\mathcal{T}}$ are sampled i.i.d from $P_{\mathcal{S}}, P_{\mathcal{T}}$ respectively where $P_{\mathcal{S}}$ is some joint distribution over $X, Y$, which take on values in $\mathcal{X}, \mathcal{Y} = [k + 1]$ and $P_{\mathcal{S}}(Y = k + 1) = 0$. $P_{\mathcal{T}}$ is a mixture distribution:

$$P_{\mathcal{T}} = (1 - \alpha)P_{\mathcal{T},[k]}(\mathbf{x}, y) + \alpha P_{\mathcal{T},k+1}(\mathbf{x}). \tag{1}$$

Here we use the notation $\alpha := P_{\mathcal{T}}(Y = k + 1)$ and $P_{\mathcal{T},k+1}(\mathbf{x}) := P_{\mathcal{T}}(X = \mathbf{x} \mid Y = k + 1)$. Our goal is to learn a model that minimizes the error of classifying the novel class $Y = k + 1$ vs. all of the $[k]$ classes observed in the training data, where loss is calculated w.r.t $P_{\mathcal{T}}$. That is, we wish to learn a classifier $h : \mathcal{X} \to [k + 1]$ that minimizes the following risk

$$\mathcal{R}_{\mathcal{T}}^{l_{01}}(h) = \mathbb{E}_{y,\mathbf{x} \sim P_{\mathcal{T}}}[h(\mathbf{x}) \neq y] = (1 - \alpha)\mathcal{L}_{P_{\mathcal{T},[k]}}^{01}(h) + \alpha \cdot \mathcal{L}_{P_{\mathcal{T},k+1}}^{01}(h), \tag{2}$$

where we denoted the accuracy of a classifier $h$ w.r.t to distribution $P$ by $\mathcal{L}_P(h)$. For simplicity, we treat the loss on novel instances the same as losses on known classes, and define the average $0 - 1$ loss on all examples. However in practice, for instance when novelty detection is required for safety purposes, we might place more weight on detecting novelties and use different metrics. Indeed in our empirical study, we will also present metrics for the novelty detection part alone. Therefore for the derived binary classifier, $h_{\text{novel}}(\mathbf{x}) = \mathbf{1}[h(\mathbf{x}) = k + 1]$ aimed at detecting novel instances, we also denote the risk w.r.t. to the novel class alone by $\mathcal{R}_{\text{novel}}^{l_{01}} = \mathbb{E}_{y,\mathbf{x} \sim P_{\mathcal{T}}}[h_{\text{novel}}(\mathbf{x}) \neq \mathbf{1}[y = k + 1]]$.

The two components setting this problem apart from standard supervised learning is the presence of a novel class (hence we emphasized this by writing the losses separately in eq. (2)), and that training data is sampled from $P_{\mathcal{S}}$, i.e. it is a domain adaptation problem. Hence it is called Open Set Domain Adaptation (OSDA) as defined formally below.

**Definition 1.** *An Open Set Domain Adaptation problem with hypothesis class $\mathcal{H}$ is a tuple $\langle P_{\mathcal{S}}(\mathbf{x}, y), P_{\mathcal{T},[k]}(\mathbf{x}), P_{\mathcal{T},k+1}, \alpha, n_{\mathcal{S}}, n_{\mathcal{T}} \rangle$, where we are given $n_{\mathcal{S}}$ and $n_{\mathcal{T}}$ i.i.d examples from $P_{\mathcal{S}}$ and $P_{\mathcal{T}}$ (defined in eq. (1)) respectively. We denote the minimizer of $\mathcal{R}_{\mathcal{T}}^{l_{01}}$ by $h^* \in \mathcal{H}$, and that of the novel class risk $\mathcal{R}_{\text{novel}}^{l_{01}}$ by $h_{\text{novel}}^*$. We further define $\beta(h) = \mathbb{E}_{\mathbf{x} \sim S_{\mathcal{S}}}[h_{\text{novel}}], \alpha(h) = \mathbb{E}_{\mathbf{x} \sim S_{\mathcal{T}}}[h_{\text{novel}}]$ as the False Positive Rate (FPR) and recall respectively, of a novelty detector derived from $h \in \mathcal{H}$.*

### 3.1 NECESSARY AND SUFFICIENT ASSUMPTIONS FOR OSDA UNDER BACKGROUND SHIFT

Without any assumptions on the relation between $P_{\mathcal{S}}$ and $P_{\mathcal{T}}$ it is impossible to obtain a guarantee for better-than-chance accuracy, even if $\mathcal{R}_{\mathcal{T}}^{l_{01}}(h^*) = 0$ for some hypothesis $h^*$ [2].
**Background shift.** We define any distribution shift between $P_{\mathcal{S}}$ and $P_{\mathcal{T}}$ that maintains support overlap such that $\text{Supp}(P_{\mathcal{T},[k]}) \subseteq \text{Supp}(P_{\mathcal{S}})$ as background shift.
Using this definition, we rely on the following separability assumption.

**Assumption 1** (separability). *There exists $h^* \in \mathcal{H}$ such that $\mathcal{R}_{\text{novel}}^{l_{01}}(h^*) = 0$. Furthermore, it holds that $\text{Supp}(P_{\mathcal{T},[k]}) \subseteq \text{Supp}(P_{\mathcal{S}})$ i.e. background shift exists between $P_{\mathcal{S}}$ and $P_{\mathcal{T},[k]}$*

The assumption states that there exists a hypothesis $h^*$ which perfectly distinguishes the novel class from the non-novel ones, and also that the support of the non-novel classes in $P_{\mathcal{T}}$ is contained in their support under $P_{\mathcal{S}}$. The second part of the assumption is intuitive, as it is makes sense to consider instances which exceed the support of $P_{\mathcal{S}}$ as novelties. The first part of the assumption, which assumes perfect classification via $h^*$ is also rather intuitive, and holds at least approximately for many problems we consider in novelty detection, such as detection of novel semantic visual concepts. Let us emphasize two aspects of our assumption.

**Separability and background shift characteristics in known classes.** The separability assumption is required only for the novel class $Y = k + 1$ vs. the known ones, we do not explicitly limit the $k$ known classes to be separable amongst themselves. Such an assumption would have placed the

---

[2]see Prop. 1 in Garg et al. (2022), Prop. 2.1 in Wald & Saria (2023), or discussion in Bekker et al. (2019) for results of this flavor

shift between $P_S$ and $P_{\mathcal{T},[k]}$ purely in the realm of covariate shift (Shimodaira, 2000). However, background shift with assumption assumption 1 facilitates forms of label shift and covariate shift thereby allowing for more general shifts. In our experiments we mostly create distribution shifts such that $P_{\mathcal{T},[k]}(X|Y) \neq P_S(X \mid Y)$ which follow the definition of background shift.

**Insufficiency of less stringent assumptions.** While generalization bounds for domain adaptation are well known from seminal works such as Ben-David et al. (2010), they are less common in the Open Set learning literature, hence let us focus on this aspect of our problem, i.e. detecting the novel class. To the best of our knowledge the only characterized sufficient and necessary conditions for (non-separable) OSDA are those in Garg et al. (2022), which *hold only for label shift*, where it is assumed that $P_S(X \mid Y = y) = P_{\mathcal{T}}(X \mid Y = y)$ for all $y \in [k]$. For this special case, they propose two assumptions which are sufficient (when added on top of the label-shift assumption) to guarantee $h^*$ can be learned from observed data. Their first assumption is *(Strong Positivity)*: there exists $X_{sep} \in \mathcal{X}$ such that $P_{\mathcal{T},k+1}(X_{sep}) = 0$ and the matrix $[P_S(\mathbf{x} \mid y)]_{\mathbf{x} \in X_{sep}, y \in [k]}$ is full rank and diagonal. We show that once more general shifts than label shifts are allowed, e.g. background/covariate shift, this condition is no longer sufficient and in fact no algorithm can guarantee better-than-chance detection. The proof for this is in A.1.

**Lemma 1.** *Let $\mathcal{A}$ be an algorithm for Open-Set Domain Adaptation. There are distributions $P_S, P_{\mathcal{T},[k]}, P_{\mathcal{T},k+1}$ such that the problem satisfies strong positivity, and $\exists h^* \in \mathcal{H}$ for which $R_{\mathcal{T}}^{l01}(h^*) = 0$, while $\mathbb{E}_{S_S, S_{\mathcal{T}}}\left[R_{\mathcal{T}}^{l01}(\mathcal{A}(S_S, S_{\mathcal{T}}))\right] \geq 0.5$.*

The second assumption proposed in Garg et al. (2022) is indeed separability as defined above; however, since their assumption is combined with the label shift assumption, the methods they develop are tailored to that scenario and do not apply to our problem. Note that Lemma 1 is stronger than Proposition 3.1 in Wald & Saria (2023). Both are impossibility statements, but Lemma 1 in our work shows impossibility under an additional assumption of strong positivity. This is crucial, since Garg et al. (2022) gives guarantees on OSDA with label shift under this strong positivity assumption, but our lemma shows that this is impossible under the background shifts we consider.

A guarantee on the accuracy of classifying the $k$ known classes is required. Theorem 1 gives a guarantee on classifying the novel class. The guarantee on the remaining $k$ classes can be obtained from classical results in Domain Adaptation, e.g. [3] or others in the survey of Redko et al. [4]. These results show how the expected error of a classifier over a source distribution can be related to its expected error over a target distribution. Our method and its formal guarantees rely on the results of Wald & Saria (2023), which we restate here reduced to the special case of separable novel classes, for better clarity on the motivation for the learning rule.

**Theorem 1.** *[(Wald & Saria, 2023)] Let $\langle P_S, P_{\mathcal{T},[k]}, P_{\mathcal{T},k+1}, \alpha, n_S, n_{\mathcal{T}} \rangle$ define an Open-Set Domain Adaptation problem, assume assumption 1 holds. Denote $R_{n,P}(\mathcal{H})$ as the Rademacher complexity of $\mathcal{H}$ w.r.t $P$ with sample size $n$, and let $\delta > 0$. Consider $\hat{h} : \mathcal{X} \rightarrow \{0,1\}$ that solves the empirical learning rule,*

$$\max_{h \in \mathcal{H}} \hat{\alpha}(h) \ s.t. \ \hat{\beta}(h) \leq R_{n_S, P_S}(\mathcal{H})/2 + \sqrt{\ln(1/\delta)/2n_S}, \tag{3}$$

*where $\hat{\alpha}(h), \hat{\beta}(h)$ are empirical estimates of $\alpha(h), \beta(h)$ from $S_{\mathcal{T}}, S_S$. We have w.p. at least $1 - 4\delta$,*
$$R_{\text{novel}}^{l01}(\hat{h}) \leq R_{\text{novel}}^{l01}(h^*) + 2R_{n_S, P_S}(\mathcal{H}) + R_{n_{\mathcal{T}}, P_{\mathcal{T}}}(\mathcal{H}) + \sqrt{2\ln(1/\delta)}\left[2/\sqrt{n_S} + 1/\sqrt{n_{\mathcal{T}}}\right].$$

# 4 SOLUTIONS TO OSDA UNDER BACKGROUND SHIFT

## 4.1 EFFICIENT ARCHITECTURE FOR ESTIMATING THE NOVEL CLASS RATIO ($\alpha$)

To solve the eq. (3), we follow Wald & Saria (2023); Chamon et al. (2022); Cotter et al. (2019) and solve a Lagrangian optimization problem obtained by switching the role of maximization and constraints in eq. (3) and taking its dual, see eq. (4) for the full objective. However, a naïve implementation of this objective has a significant drawback in terms of computational complexity due to hyperparameter optimization. Indeed, prior work on the rate-constrained learning problems we seek to solve (where rate in our problem corresponds to the size of the novel class) is limited to either very small models and datasets (Wald & Saria, 2023), or to certain applications such as fairness (Zafar et al., 2019), where the desired rate is known a-priori.

The computationally challenging part is that our Lagrangian problem needs to be solved for each candidate value $\hat{\alpha} \in \boldsymbol{\alpha}$, corresponding to the estimate of our method to the size of the novel class, where $\boldsymbol{\alpha} \in [0,1]^L$ is a grid of size $L$ that we search over. After the grid search, the chosen model is the one obtaining the largest value $\hat{\alpha}$, while still satisfying the constraint on $\hat{\beta}(h)$ (empirical estimate of FPR i.e. false positive rate of our method while distinguishing between $P_S$ and $P_T$) specified in the learning objective. For empirical purposes, we calculate the approximate bound for $\beta$ (constraint on $\hat{\beta}(h)$) using the Rademacher complexity in theorem 1. We find that setting $\beta = 0.01$ is well within the theoretically calculated bounds and works well in practice for all the experiments. Plots in figure 3 provide further insights on the impact of varying $\beta$ over the OSDA performance of CoLOR.

To this end we train an architecture $h(\mathbf{x}) = \phi \circ w(\mathbf{x})$ where $\phi : \mathcal{X} \to \mathbb{R}^d$ is a shared representation for several heads corresponding to each candidate value $\hat{\alpha}$ on search range $\boldsymbol{\alpha}$. We denote each head by $w^\alpha : \mathbb{R}^d \to \mathcal{R}$. Hence the solution amortizes training time for all candidate values by solving the primal dual optimization problem once. which leads to better performance with large data & models in practice as we see in section 5.

## 4.2 A SIMPLE EXTENSION OF CONSTRAINED LEARNING FOR OSDA

To account for the known classes $Y = 1, \ldots, k$, we examine whether including another classification head, i.e. extending $w = [w^c, w^\alpha]$, such that $w : \mathbb{R}^d \to \mathbb{R}^{L+k}$ in the model $h(\mathbf{x}) = \phi \circ w(\mathbf{x})$, and $w^\alpha$ is fitted on training data $S_S$, is preferable to disjoint training of a novelty detector. Note that this addition of multiple classification heads for novelty detection ($[w_1^\alpha], w_2^\alpha, ..., w_L^\alpha]$) does not significantly affect the computation time w.r.t. model with single novelty detection head since the added number of parameters is not large w.r.t. rest of the network. At first sight, it is unclear whether such a simplistic approach that learns a novelty detector and classifier of known classes jointly, should be helpful beyond training each one separately. $S_S$ is known prior to training the novelty detector, and its training objective does not take into account the test data $S_T$ in any way.

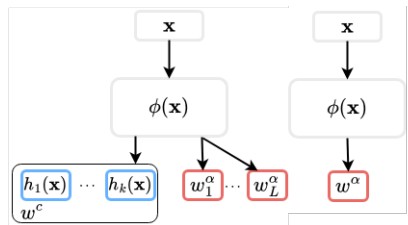

Figure 1: (left) Architecture of CoLOR for OSDA, heads for multiple recall values = $w_i^a$, and a classification head $w^c$, vs. (right) a network optimizing for novelty detection with single recall value as in Wald & Saria (2023).

Our conjecture is that by training a shared representation for both tasks, the model is learning a simpler representation that fits both tasks, yielding favorable generalization bounds as suggested by theory on multitask learning (Baxter, 2000; Maurer et al., 2016). However, the formulation of this theory has some differences from ours as it does not include novelty detection nor constrained objectives. Understanding when known theory extends to settings beyond solving multiple Empirical Risk Minimization tasks is an interesting goal for future research.

Let us gather the components for our overall method, CoLOR, that we evaluate in the next section. Figure 1 illustrates the architecture of CoLOR. It is trained by combining the constrained learning rule with $\ell_{\log}$ the binary cross-entropy loss aiming to classify examples in $S_T$ as novel,[3] $\lambda$ is a Lagrange multiplier and $\ell_\sigma(\cdot)$ is a sigmoid function which serves as an approximation to the indicator function, and the supervised cross entropy loss is $l_{ce}$. $h_c(\mathbf{x}_i) = \phi \circ w^c(x)$ represents the head of the predicted known class in $w^c$.

$$\mathcal{L}(h) = n_S^{-1} \sum_{i \in S_S} \left( l_{ce}(h_c(\mathbf{x}_i), y_i) + \sum_{\hat{\alpha} \in \boldsymbol{\alpha}} l_{\log}(h_{\hat{\alpha}}(\mathbf{x}_i), 0) \right) + \lambda \cdot n_T^{-1} \sum_{\substack{i \in S_T, \\ \hat{\alpha} \in \boldsymbol{\alpha}}} \left( l_\sigma(h_{\hat{\alpha}}(\mathbf{x}_i)) - \hat{\alpha} \right) \quad (4)$$

In our experiments, the representation $\phi(\mathbf{x})$ is either trained from scratch or uses a pretrained architecture over which we add fully connected layers. For small-scale datasets like CIFAR100 we train $\phi(x) = $ ResNet18 He et al. (2016) from scratch for the adaptive methods. However, for larger more complex datasets like SUN397, we use a pretrained $\phi(x)$ like ResNet50 pretrained on ImageNet1K_V1 Russakovsky et al. (2015) and ViT-L/14 visual encoder pretrained using CLIP Radford et al. (2021). For text datasets like Amazon Reviews, we use RoBERTa Liu et al. (2019) to

---

[3]label 0 in the loss corresponds here to a sample belonging to a known class.

obtain text embeddings. We train the hidden fully connected layer and the following classification layer required to learn shared representations and generate output logits. We use the default model for all other baselines.

---

**Algorithm 1** CoLOR: Constrained Learning for Open-set Recognition

---

**Require:** Labelled $S_{\mathcal{S}}$ and unlabelled $S_{\mathcal{T}}$ datasets, hypothesis class $\mathcal{H}$, target FPR $\beta > 0$ and potential novel class sizes $\boldsymbol{\alpha} \in [0,1]^L$.

1: Split $S_{\mathcal{S}}, S_{\mathcal{T}}$ into train $T_{\mathcal{S}}, T_{\mathcal{T}}$ and validation sets $V_{\mathcal{S}}, V_{\mathcal{T}}$ respectively.
2: Train either the entire model ($\phi$, $w^c$ & $w^\alpha$) or use pretrained $\phi$ and train just the last fully connected layer of $\phi$ along with $w^c$ & $w^\alpha_{\hat{\alpha}}$ to minimize the eq. 4.
3: Let $\hat{\beta}(w^\alpha_{\hat{\alpha}}) = \frac{1}{|V_{\mathcal{S}}|} \sum_{\mathbf{x} \in V_{\mathcal{S}}} w^\alpha_{\hat{\alpha}} \circ \phi(\mathbf{x})$, and $\hat{\alpha}(w^\alpha_{\hat{\alpha}}) = \frac{1}{|V_{\mathcal{T}}|} \sum_{\mathbf{x} \in V_{\mathcal{T}}} w^\alpha_{\hat{\alpha}} \circ \phi(\mathbf{x}) \, \forall \alpha \in \boldsymbol{\alpha}$
4: **return** $\text{argmax}_{w^\alpha_{\hat{\alpha}} : \alpha \in \boldsymbol{\alpha}, \hat{\beta}(w^\alpha_{\hat{\alpha}}) < \beta} \hat{\alpha}(w^\alpha_{\hat{\alpha}})$

---

## 5 EXPERIMENTS

We are now in place to empirically evaluate CoLOR against variety of baselines across background shifts and novel classes that we create in real data. Here, the main questions we wish to answer are:

1. Does background shift within non-novel instances affect OSDA performance?

2. Can shared representations mitigate the impact of background shift by enhancing robustness in classifying samples from $S_{\mathcal{T},[k]}$ thereby improving overall OSDA performance?

3. What is the effect of novel class ratio $\alpha$ particularly w.r.t. detecting novel classes?

### 5.1 EXPERIMENTAL SETTING

The experiments to examine these questions are devised over image and text datasets as follows. We randomly draw a class from the set of classes $\mathcal{Y}$ and assign that as the $k+1$-th novel class. Denoting the instances of known classes by $S_k = \{\mathbf{x} : y \in [k]\} \, \forall (\mathbf{x}, y) \in S$ and novel ones as $S_{k+1} = \{\mathbf{x} : y = k+1\} \, \forall (\mathbf{x}, y) \in S$, we create a background shift by further splitting $S_k$ into $S_{\mathcal{S}}$ and $S_{\mathcal{T},[k]}$. We use semantic attributes that are annotated in the metadata of each dataset to create this shift between $S_{\mathcal{S}}$ and $S_{\mathcal{T},[k]}$. The attributes used for each dataset are specified in section 5.2 and elaborated further in section A.3.1. At training time the learner is provided with a labelled dataset $S_{\mathcal{S}}$ and the unlabelled dataset $S_{\mathcal{T}} = S_{\mathcal{T},[k]} \cup S_{k+1}$. The mixture proportion $\alpha = |S_{k+1}|/(|S_{k+1}| + |S_{\mathcal{T},[k]}|)$ is set by adjusting the sizes of the selected novel classes in $S_{\mathcal{T},[k]}$.

### 5.2 DATASETS

Most of the large scale image classification models are trained on ImageNet dataset. Hence we employ a similar large scale dataset called SUN397 Xiao et al. (2010) having completely different categories than ImageNet. We also include a Amazon Reviews text dataset Ni et al. (2019) to demonstrate the versatility of the method across multiple modalities. Finally, we include small scale CIFAR100 Krizhevsky (2009) dataset to evaluate the adaptive methods (like DD, nnPU, uPU, BODA, OSDA) particularly in scenarios when their feature extractors are trained from scratch. Table table 1 provides a summary of characteristics and further details on each dataset are in appendix A.3.1.

- **SUN397.** Popular scene understanding dataset with a three-level hierarchy of scene categories Xiao et al. (2010), exploiting this structure to create distribution shifts by varying subtype ($y$) proportions within primary class labels $\mathcal{Y}$. Indoor scenes (e.g., shopping/dining places, workplaces) serve as known (in-distribution) classes, while outdoor scenes are randomly selected as novel classes. Varying subtype proportions in indoor categories causes a background shift since $\text{Supp}(P_{\mathcal{T},[k]}) \subseteq \text{Supp}(P_{\mathcal{S}})$.

- **Amazon Product Reviews.** Classes are different product categories (prime pantry, musical instruments, etc.), and induce background/covariate shift in known classes based on positive/negative sentiment in the review. The novel class is an unknown product category.

- **CIFAR100.** Similar to the SUN397 dataset, we leverage the inherent hierarchies of CIFAR100 classes to create a natural background shift based on subtypes (e.g. dolphins) of known primary

categories like aquatic mammals. Novel classes are chosen from other categories in distinct branches of the class hierarchy.

Table 1: Overview of experiment settings. DS = distribution shift, prop. = proportions

| Experiment setup | SUN397 | CIFAR100 | Amazon Reviews |
|---|---|---|---|
| DS factor | varing subtypes prop. | varing subtypes prop. | sentiment |
| no. of novel classes | 12 | 5 | 6 |
| Novel class ratio ($\alpha$) | $0.07 \pm 0.03$ | $0.16 \pm 0.09$ | $0.07 \pm 0.02$ |

**Evaluation metrics:** We primarily use Area Under ROC Curve (AUROC) and Area Under Precision-Recall Curve (AUPRC) to evaluate the novel category detection performance, while we use Open-Set Classification Rate (OSCR) to summarize overall OSDA performance for all the methods [4]. OSCR measures the trade-off between correct classification rate of the known classes and false positive rate of the novel samples, we refer to Dhamija et al. (2018) for details of the metric.

## 5.3 BASELINE METHODS

We include adaptive methods from novelty detection that access both labelled source and unlabelled target data. These baselines are domain discriminator (DD), Elkan & Noto (2008); du Plessis et al. (2014); Garg et al. (2021), uPU du Plessis et al. (2014) and nnPU Kiryo et al. (2017). These methods are modified for OSDA through joint learning approach enabled by a simple architectural modification like figure 1 and aggregating the loss components from both the tasks. We also include another popular OSDA baseline BODA (Saito et al., 2018) that is agnostic to data modality. Another adaptive baseline is Li et al. (2023), however, this is specifically designed for vision data.

We include an entropy-based method, ARPL (Chen et al., 2021) with Maximum Logit Score as proposed in (Vaze et al., 2022). Such methods are non-adaptive as they do not access target data, yet are a popular choice for novelty detection.[5] We also tested simple and popular baselines such as MSP Hendrycks & Gimpel (2016) and results can be found in appendix tables 13, 14. We implemented the SHOT method from Liang et al. (2021), but found that the clustering approach performed worse than MSP or entropy-based methods. We further include a zero-shot OOD detection method (ZOC) proposed in Esmaeilpour et al. (2022). Further details about training and hyperparameters are in Appendix A.3.2.

Table 2: OSDA under background shift performance comparison of the adaptive methods that are versatile across different data modalities. Detailed results are in appendix tables 5, 8, 10).[6]

| Metric | Methods | SUN397 ($\alpha = 0.07 \pm 0.03$) | | CIFAR100 ($\alpha = 0.07 \pm 0.02$) | | Amazon Reviews ($\alpha = 0.16 \pm 0.09$) | |
|---|---|---|---|---|---|---|---|
| | | Summary | Wins | Summary | Wins | Summary | Wins |
| AUROC | DD | $0.91 \pm 0.05$ | 1/15 | $0.70 \pm 0.11$ | 6/25 | $0.72 \pm 0.09$ | 1/30 |
| | uPU | $0.76 \pm 0.14$ | 0/15 | $0.67 \pm 0.10$ | 1/25 | $0.76 \pm 0.08$ | 6/30 |
| | nnPU | $0.76 \pm 0.14$ | 0/15 | $0.67 \pm 0.12$ | 1/25 | $0.76 \pm 0.08$ | 6/30 |
| | BODA | $0.86 \pm 0.06$ | 0/15 | $0.57 \pm 0.06$ | 1/25 | $0.66 \pm 0.10$ | 5/30 |
| | CoLOR | $\mathbf{0.98 \pm 0.02}$ | $\mathbf{14/15}$ | $\mathbf{0.77 \pm 0.09}$ | $\mathbf{16/25}$ | $\mathbf{0.79 \pm 0.09}$ | $\mathbf{18/30}$ |
| AUPRC | DD | $0.54 \pm 0.22$ | 1/15 | $0.24 \pm 0.13$ | 3/25 | $0.43 \pm 0.18$ | 0/30 |
| | uPU | $0.21 \pm 0.22$ | 0/15 | $0.18 \pm 0.12$ | 1/25 | $0.51 \pm 0.17$ | 7/30 |
| | nnPU | $0.21 \pm 0.22$ | 0/15 | $0.20 \pm 0.16$ | 2/25 | $0.51 \pm 0.17$ | 7/30 |
| | BODA | $0.45 \pm 0.11$ | 0/15 | $0.09 \pm 0.03$ | 1/25 | $0.28 \pm 0.21$ | 2/30 |
| | CoLOR | $\mathbf{0.91 \pm 0.09}$ | $\mathbf{14/15}$ | $\mathbf{0.33 \pm 0.14}$ | $\mathbf{18/25}$ | $\mathbf{0.54 \pm 0.18}$ | $\mathbf{21/30}$ |
| OSCR | DD | $0.68 \pm 0.05$ | 1/15 | $0.51 \pm 0.10$ | 2/25 | $0.50 \pm 0.06$ | 0/30 |
| | uPU | $0.40 \pm 0.11$ | 0/15 | $0.49 \pm 0.09$ | 0/25 | $0.54 \pm 0.05$ | 6/30 |
| | nnPU | $0.40 \pm 0.11$ | 0/15 | $0.48 \pm 0.10$ | 1/25 | $0.54 \pm 0.05$ | 6/30 |
| | BODA | $0.55 \pm 0.10$ | 0/15 | $\mathbf{0.60 \pm 0.04}$ | 10/25 | $\mathbf{0.56 \pm 0.06}$ | $\mathbf{13/30}$ |
| | CoLOR | $\mathbf{0.81 \pm 0.04}$ | $\mathbf{14/15}$ | $0.59 \pm 0.08$ | $\mathbf{12/25}$ | $\mathbf{0.56 \pm 0.06}$ | 11/30 |

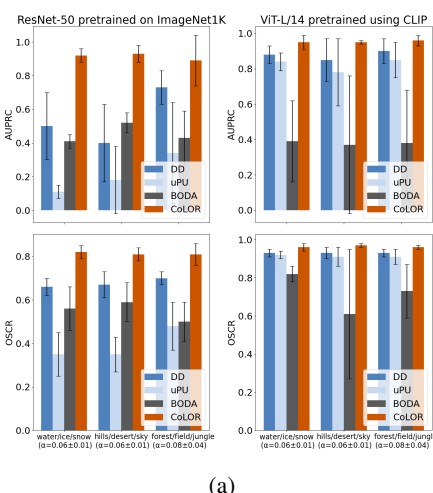 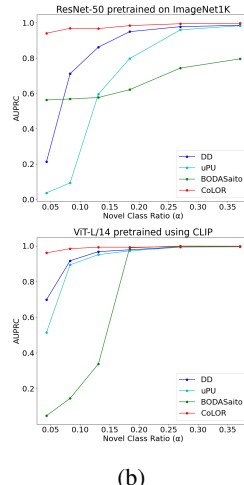

|  |  |
|---|---|
| (a) | (b) |

Figure 2: (a) OSDA performance of top performing adaptive methods on SUN397 dataset with background shift using pretrained ResNet50 & CLIP ViT-L/14 backbone architectures. (b) Impact of novel class ratio ($\alpha$) on adaptive methods on SUN397 dataset under background shift.

## 5.4 RESULTS

Table 2 summarizes the OSDA performance across all the datasets using all the metrics discussed previously and we observe that CoLOR generally outperforms all the baselines with a significant margin. The average OSCR scores/no. of wins of CoLOR is comparable to BODA for CIFAR100 and Amazon Reviews dataset however, BODA notably under performs in novelty detection for the same datasets. Additionally, we observe that BODA does not perform as well on the larger dataset of SUN397. We find that the nnPU objective defaults to uPU when using pretrained feature extractors, as the empirical risk remains non-negative. This causes similar results of uPU and nnPU on SUN397 and CIFAR100 datasets. In the image datasets, we find that adaptive methods significantly outperform non-adaptive methods overall. We further strengthen our results by extending the study to other richer feature representations like pretrained CLIP ViT-L/14, particularly for SUN397 dataset. This is shown in 2a across different novel classes (X-axis) of SUN397 dataset. CoLOR still outperforms the baselines using either of the backbone architectures. We can see that CLIP ViT-L/14 have overall better performance than ResNet50 due to richer features. Refer tables 5, 6, 8, 10 in appendix for further results.

## 5.5 DISCUSSION

Based on our observations from the experiments, we address the three critical questions below:

**Background shift within non-novel instances causes a significant drop in the OSDA performance of all the methods that involve fine-tuning using the source data.**
From tables 3b, 6 and 7 we observe that all the methods that involve training a closed-set classifier suffer from background shift. Hence, background shift is harming the the performance of the closed set classifier and the open set detector. If we had a better closed set classifier, in the sense that it was more robust to the background shift, then we probably would've ended up with better OSDA performance too. But CoLOR seems to primarily help when standard techniques like long training, label smoothing, and other techniques used in Vaze et al. are insufficient to train a model that's robust to shift. This is not the case for ZOC as it performs zero-shot open-set classification using pretrained

---

[4]Section A.2.1 in appendix provides detailed argument for the preference of AUPRC over AUROC mainly when minority class proportions are very low.

[5]We are using ARPL on SUN397. The paper that proposed ARPL also introduced an improvement with Confused Sampling (ARPL+CS) that uses GANs, but that proved challenging to apply in the SUN397 dataset

[6]The summary statistics are derived by averaging (or aggregating wins) across all novel class identities and randomly generated data splits (between $P_{\mathcal{S}}$ & $P_{\mathcal{T}}$) along with the corresponding standard deviations. Refer A.3.2

Table 3: (a) Top-1 accuracy over $S_{\mathcal{T}}$ for closed-set classification on SUN397 data. (b) Adaptive & non-adaptive baselines with (w/) and without (w/o) background shift (DS) performance on SUN397.

| Method | ResNet50 | ViT-L/14 from CLIP |
|---|---|---|
| Source-only | $0.72 \pm 0.03$ | $0.97 \pm 0.01$ |
| DD | $0.75 \pm 0.04$ | $0.97 \pm 0.01$ |
| BODA | $0.70 \pm 0.07$ | $0.75 \pm 0.13$ |
| CoLOR | $\mathbf{0.83 \pm 0.03}$ | $\mathbf{0.98 \pm 0.01}$ |

(a)

| Method | AUROC | | AUPRC | | OSCR | |
|---|---|---|---|---|---|---|
| | w/ DS | w/o DS | w/ DS | w/o DS | w/ DS | w/o DS |
| DD (ViT-L/14) | $0.96 \pm 0.02$ | $\mathbf{1.00 \pm 0.00}$ | $0.87 \pm 0.08$ | $\mathbf{1.00 \pm 0.00}$ | $0.93 \pm 0.02$ | $\mathbf{0.99 \pm 0.00}$ |
| uPU (ViT-L/14) | $0.95 \pm 0.04$ | $\mathbf{1.00 \pm 0.00}$ | $0.82 \pm 0.12$ | $\mathbf{1.00 \pm 0.00}$ | $0.92 \pm 0.03$ | $\mathbf{0.99 \pm 0.00}$ |
| BODA (ViT-L/14) | $0.84 \pm 0.15$ | $0.88 \pm 0.04$ | $0.38 \pm 0.29$ | $0.24 \pm 0.15$ | $0.72 \pm 0.21$ | $0.93 \pm 0.02$ |
| ARPL (custom default) | $0.71 \pm 0.08$ | $0.84 \pm 0.04$ | $0.12 \pm 0.07$ | $0.20 \pm 0.07$ | $0.60 \pm 0.06$ | $0.80 \pm 0.04$ |
| ZOC (custom default) | $0.82 \pm 0.07$ | $0.82 \pm 0.08$ | $0.23 \pm 0.06$ | $0.22 \pm 0.07$ | $0.51 \pm 0.06$ | $0.49 \pm 0.05$ |
| ANNA (custom default) | $0.93 \pm 0.05$ | $0.95 \pm 0.07$ | $0.73 \pm 0.16$ | $0.90 \pm 0.08$ | $0.60 \pm 0.07$ | $0.82 \pm 0.07$ |
| CoLOR (ViT-L/14) | $\mathbf{0.99 \pm 0.01}$ | $\mathbf{1.00 \pm 0.00}$ | $\mathbf{0.95 \pm 0.03}$ | $0.99 \pm 0.01$ | $\mathbf{0.96 \pm 0.01}$ | $0.98 \pm 0.00$ |

(b)

CLIP and text decoder models without accessing source and target data. We have provided further results on CIFAR100 dataset in 8 & 9 although CIFAR100 is a much simpler dataset and hence we see near perfect scores by ZOC due to the use of CLIP model pretrained on large image caption datasets. We also see from tables 6 & 7 that non-adaptive methods do not perform as well when extended to larger dataset of SUN397.

**Shared representations obtained by joint learning of closed set classification and novel category detection can help mitigate the impact of background shift within non-novel instances.**
Table 3a examines the effect of each component of the learning objective from equation 4 when there is background shift between $S_{\mathcal{S}}$ and $S_{\mathcal{T}}$. The Source-only method is trained on labelled $S_{\mathcal{S}}$ for classifying known classes in $S_{\mathcal{T}}$, while DD combines closed-set classification with domain discrimination (distinguishing samples from $S_{\mathcal{S}}$ and $S_{\mathcal{T}}$). We can see that DD has better top-1 accuracy of known classes than the Source-only method on $S_{\mathcal{T}}$. This difference is notable for ResNet50 model. CoLOR further improves performance by employing constrained learning to detect novel classes instead of domain discrimination in the joint learning objective. Performance differences are minimal for CLIP ViT-L/14 due to its already rich and robust representations.

**As the novel class ratio $\alpha$ decreases, the performance of existing methods to detect unknown/novel classes significantly decreases.** Figure 2b illustrates the effect of novel class ratio on the novel class detection performance of existing OSDA methods. A low $\alpha$ notably deteriorates the open-set classification performance, especially for methods not designed to handle such cases. Since many benchmarks focus on large novel class sizes, we believe that experimenting with smaller sizes would be a valuable step toward creating more realistic benchmarks.

# 6    FUTURE WORK

Future research could focus on refining the search criteria for the optimal model head by selecting an appropriate $\beta$ threshold, as the bounds provided in Theorem 1 are loose, like most bounds in learning theory, and may be suboptimal. Additionally, a robust theoretical underpinning is needed to explain why shared representations improve the classification of known classes under distribution shifts. Integrating test-time or model-free domain adaptation methods, such as those by Wang et al. (2020), Zhang et al. (2023), or Saito et al. (2018), could further enhance OSDA performance. It would be valuable to compare our approach with large foundation models. Although curating such a benchmark would require access to publicly available pretraining datasets like LAION, which would require significant effort and present unforeseen challenges.

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

# A  APPENDIX / SUPPLEMENTAL MATERIAL

## A.1  PROOF OF LEMMA 1

Let us recall the strong positivity assumption stated in the main paper, which appears in Garg et al. (2022).

**Assumption 2** (Strong positivity). *There exists $X_{sep} \subseteq \mathcal{X}$ such that $P_{\mathcal{T},1}(X_{sep}) = 0$ and the matrix $[P_{\mathcal{S}}(\mathbf{x} \mid y)]_{\mathbf{x} \in X_{sep} my \in [k]}$ is full rank and diagonal.*

We restate and prove the claim that Open-Set Domain Adaptation is not learnable under this assumption, once the label shift assumption is removed.

**Lemma.** *Let $\mathcal{A}$ be an algorithm for Open-Set Domain Adaptation. There are distributions $P_{\mathcal{S}}, P_{\mathcal{T},[k]}$ and $P_{\mathcal{T},k+1}$ such that the problem satisfies strong positivity, and $\exists h^* \in \mathcal{H}$ for which $R_{\mathcal{T}}^{l_{01}}(h^*) = 0$, while $\mathbb{E}_{S_{\mathcal{S}},S_{\mathcal{T}}} \left[ R_{\mathcal{T}}^{l_{01}}(\mathcal{A}(S_{\mathcal{S}}, S_{\mathcal{T}})) \right] \geq 0.5$.*

*Proof.* Define the following distributions over 4 states

$$[P_{\mathcal{S}}(x \mid y)]_{x \in \mathcal{X}, y \in [k]} = \begin{bmatrix} 1 - \varepsilon & 0 & 0 & \varepsilon \\ 0 & 1 - \varepsilon & \varepsilon & 0 \end{bmatrix},$$

$$P_{\mathcal{S}}(Y) = [\frac{1 - 2\varepsilon}{1 - \varepsilon}, \frac{\varepsilon}{1 - \varepsilon}, 0]$$

for some $\varepsilon > 0$, and two other distribution over $\mathcal{X}$, $Q(X) = [0, 0, 1, 0]$ and $D(X) = [0, 0, 0, 1]$. Consider 2 Open-Set Domain Adaptation problems where $k = 2$ and $\alpha = 0.5$:

- One where $P_{\mathcal{T},[k]} = Q$ and $P_{\mathcal{T},k+1} = D$, which means that $P_{\mathcal{T},[k]}(X \mid Y = y) = [0, 0, 1, 0]$ and we set $P_{\mathcal{T},[k]}(Y) = [\frac{1-2\varepsilon}{1-\varepsilon}, \frac{\varepsilon}{1-\varepsilon}, 0]$ although we can set it to any arbitrary distribution.

- For the second problem $P_{\mathcal{T},[k]} = D$ and $P_{\mathcal{T},k+1} = Q$, which entails similarly to the first case that $P_{\mathcal{T},[k]}(X \mid Y = y) = [0, 0, 0, 1]$ for $y \in [k]$, while we keep $P_{\mathcal{S}}$ and the rest of the details as they are in the first problem.

It is clear that under the hypothesis class $\mathcal{H}$ of all binary classifiers on $\mathcal{X}$, it holds that $R_{\mathcal{T}}^{l_{01}}(h^*) = 0$. Now we will show that both problems satisfy strong positivity (note that they also satisfy that $\text{Supp}(P_{\mathcal{T},[k]}) \subseteq \text{Supp}(P_{\mathcal{S}})$), and also $P_{\mathcal{S}}(X, Y)$ and $P_{\mathcal{T}}(X)$ are the same for both problems.

Once this is shown, we can conclude our result, since any observed dataset that is an input to $\mathcal{A}$ is equally likely in both problems. However, any hypothesis $h$ that achieves $R_{\mathcal{T}}^{l_{01}}(h) = \delta$ on the first problem, achieves risk $1 - \delta$ on the other problem since $P_{\mathcal{T},[k]}$ and $P_{\mathcal{T},k+1}$ switch roles between the two problems.

To show that the problems satisfy strong positivity, consider $X_{sep}$ as the first and second states. We have that for both problem $P_{\mathcal{T},k+1}(X_{sep} = 0$ since both $D(X_{sep}) = 0$ and $Q(X_{sep}) = 0$, while

$$[P_{\mathcal{S}}(x \mid y)]_{x \in X_{sep}, y \in [k]} = \begin{bmatrix} 1 - \varepsilon & 0 \\ 0 & 1 - \varepsilon \end{bmatrix},$$

which is a full rank and diagonal matrix. Hence the strong positivity condition is satisfied. We defined the same $P_{\mathcal{S}}(X, Y)$ for both problems, so it is left to show that $P_{\mathcal{T}}(X)$ also equals for them. This is also straightforward as for both problem $P_{\mathcal{T}}(X) = 0.5 \cdot Q + 0.5 \cdot D$, which concludes the proof.  $\square$

## A.2  ADDITIONAL DETAILS ON EXPERIMENTAL SETTING

### A.2.1  AUROC VS AUPRC SCORES FOR NOVEL CLASS DETECTION

It is a common argument in the machine learning literature that AUPRC scores are more suitable for evaluating methodologies in class-imbalanced scenarios. However, this stance is nuanced, as some research, such as McDermott et al. (2024) suggests favoring AUROC over AUPRC in certain imbalanced conditions. McDermott et al. (2024) further notes that AUPRC inherently emphasizes

the performance on samples with higher scores. Given our experimental focus on assessing models' ability to assign higher scores for novelty detection, AUPRC emerges as the most relevant metric for our analysis, especially when the proportions of positive class (novel class in our case) is very low.

### A.2.2 DISCUSSION

Building on the results and observations from the previous section, we proceed to further analyze and understand the aspects of OSDA under conditions of background shift. The curve plots in 2b compare the novelty detection performance of the methods shown , it is evident that CoLOR outperforms other methods particularly when the novel class ratio $\alpha$ is less than 0.2. However, as $\alpha$ is increased beyond 0.3, the other baselines rapidly catch up to the AUPRC performance of CoLOR. From Table 3a we observe that using constrained learning to acquire shared representations benefits the classification performance on known classes. The source-only method mentioned in 3a serves as a baseline, trained exclusively on the $S_{\mathcal{S}}$ and evaluated on the $S_{\mathcal{T}}$ without employing any strategies to mitigate shift effects. Furthermore, Table 3b provides insights on the impact of distribution shift on the overall OSDA performance of all the methods using ViT-L/14 visual encoder pretraiend using CLIP.

We specify our empirical test of measuring separability in A.3.1. Based on this test of separability, we observe that novel classes in Amazon Reviews dataset are not perfectly separable and hence violate the assumption of separability to some extent. However, we see in our result in tables 2, 10 and 12 that CoLOR still outperforms other baselines. Hence, we realize that CoLOR is robust to certain violations of the separability assumption.

### A.2.3 IMPLEMENTATION OF CoLOR

It is important to note that each model head $h_{\hat{\alpha}}$ independently solve constrained problem in 4 using primal-dual optimization while remaining $k$ heads focus on classifying samples (from $S_{\mathcal{S}}$) from $k$ known categories using ground truths $\mathcal{Y}_{\mathcal{S}}$.

## A.3 EXTENDING CONSTRAINED LEARNING OBJECTIVE TO OSDA

We either train an entire model (encoder + classifier) from scratch or just add two fully-connected (FC) layers on top of a pretrained encoder and only train these additional FC layers. The first FC layer provides the shared representation acquired through learning from the related tasks while second FC layer uses this shared representation to classify the known classes and detect novel identities.

### A.3.1 DATASET

One important factor to consider while creating shifts is that $S_{\mathcal{S}}$ and $S_{\mathcal{T},k+1}$ should be distinguishable. As dataset separability is a difficult quantity to measure we train a classifier (oracle) for each dataset to distinguish novel groups from samples belonging to known categories. We then use the learnability of the oracle as a criterion to ensure the separability of novel classes. This means that we calculate the AUROC and AUPRC scores of the oracle for the task of supervised novelty detection. Higher AUROC and AUPRC would correspond to higher separability. We consider an AUROC and AUPRC higher than 0.98 as ideal to ensure separability between novel class identities and known classes. It is difficult to ensure such high separability for all the datasets, particularly for Amazon Reviews dataset which does not perfectly satisfy the separability. Yet we observe that CoLOR is robust to background shift in such settings outperforming all the baselines as observed in Tables 2, 10 and 12.
We conduct 5 repetitions of an experiment for each dataset and for every identity of the novel class. Each repetition uses a unique random seed, representing a distinct background shift setting. These settings are generated by randomly varying the subtype proportions of the known categories within the SUN397 dataset.

**SUN397:** It consists of images of scenes/places from various locations. The dataset is provided with 3 levels of hierarchy where Level-1 is grouped as indoor, natural outdoor and man-made outdoor scenes. We use indoor classes as in-distribution classes while we choose novel classes from natural outdoor scenes. Level-2 hierarchy has shopping, workplace, homes/hotels, etc. under indoor category while outdoor natural contains classes like water/ice/snow, mountains/hills desert/sky, forest, etc. Each of these level-2 categories further has subcategories (level-3 classes) that form these

level-2 groupings. We randomly select 8 level-3 subtypes (like bakery shop or banquet hall) per level-2 category (shopping/dining places) and vary the subtype proportions to create background shift between source and target. Furthermore, novel classes are randomly selected from the level-2 categories of outdoor natural group.

**CIFAR100:** It consists of 60,000 32x32 colour images in 20 primary classes (superclasses) each of which have 5 subcategories composing a total of 100 classes Krizhevsky et al. (2009). The training set has 50,000 images (i.e. 500 images per subcategory and 2500 images per primary class) while the test set has 10000 images (i.e. 100 images per subcategory and 500 images per primary class). We retain these splits for our experiments. We use 4 primary categories as aquatic mammals, flowers, fishes and birds. The subcategories of the aquatic mammals are beaver, dolphin, otter, seal are whale while that of flowers are orchid, poppy, rose, sunflower and tulip. Similarly fishes and birds have 5 subcategories each. We vary the marginal distribution of these subcategories to create a subpopulation shift leading to a background shift between source and target data while maintaining no label shift w.r.t. primary categories i.e. aquatic mammals and flowers. The novel category is randomly selected from the remaining unseen categories.

**Amazon Reviews** The dataset is heavily skewed with respect to sentiments and product categories. Hence, we select 6 product categories having similar orders of magnitude of the number of reviews namely 'Digital Music', 'Industrial & Scientific', 'Luxury Beauty', 'Musical Intstruments', 'Prime Pantry' and 'Software'. To prevent further skewness in the dataset due to sentiments, we restrict the sample size per sentiment per category to 500 reviews in the training set and 125 reviews per sentiment per product category. We induce a background shift based on sentiments. Reviews with rating strictly below 3.0 (out of 5.0) are considered negative sentiments and those with rating strictly above 3.0 are considered positive sentiments whereas reviews having a rating of exactly 3.0 are discarded from the dataset. The minimum rating is 1.0 while the maximum is 5.0.

### A.3.2 HYPER-PARAMETERS AND TRAINING

For all the methods, we keep the set of candidate target recall values consistent through all the experiments $\alpha = [0.02, 0.05, 0.10, 0.15, 0.20, 0.25, 0.30, 0.35, 0.40, 0.45]$. We consistently set the FPR threshold $\beta = 0.01$ without optimizing it at all based on validation dataset. For each novel class identity, we repeat the experiments for 5 different randomly generated splits between $P_S$ and $P_T$ adhering to the definition and assumptions of background shift. For CIFAR100, we use ResNet18 backbone followed by a linear layer for classification and train the whole model from scratch for all the methods. For Amazon Reviews dataset, we use pretrained RoBERTa features followed by 2 linear layers for classification. For Amazon Reviews, due to computational limitations we resort to linear probing rather than finetuning the whole model and only finetune the last two linear layers. The primary hyperparameters we tune for stable convergence are learning rate, L2 weight penalty scalar to prevent overfitting, logit multiplier values that act as temperature controllers for softmax/sigmoid scores and gradient clipping to avoid exploding gradients. These hyper parameters are tuned based on a sample training and validation sets but are kept constant throughout the dataset and baseline across different seed values and novelty cases. Furthermore, the methods CoNoC & CoLOR require additional hyperparameters like dual learning rate and lagrange multipliers. Each output node is associated with Lagrange multipliers, which address a distinct primal-dual optimization problem owing to diverse target recall constraints. These multipliers are initialized to 1.0, while the dual learning rate is meticulously calibrated for CIFAR100 & Amazon Reviews datasets individually to ensure stable learning dynamics conducive to minimizing both the objective surrogate loss function for FPR and recall inequality constraints. Note that for ZOC, We used cosine similarity between image embeddings and the known class text embeddings to obtain the closed-set class predictions. Table 4 displays all the hyperparameters used for each of the baselines and datasets.

### A.4 PERFORMANCE COMPARISON BASED ON AVERAGE RELATIVE & ABSOLUTE AU-ROC AND AU-PRC SCORES

Refer tables 8, 10, 5, 6, 7, 3b, 3a below.

Table 4: Hyperparameters: lr = learning rate, dlr = dual learning rate (CoLOR), L2 penalty = L2 weight penalty scaler, lm = logit multiplier, clip = gradient clipping value. The two values separated by "/" in learning rate column of SUN397 dataset correspond to the linear probing of ResNet50 (pretrained on ImageNet) and ViT (pretrained using CLIP) backbones respectively.

| Method | CIFAR100 | | | | | Amazon Reviews | | | | | SUN397 | | | | |
|---|---|---|---|---|---|---|---|---|---|---|---|---|---|---|---|
| | lr | dlr | L2 penalty | lm | clip | lr | dlr | L2 penalty | lm | clip | lr | dlr | L2 penalty | lm | clip |
| DD | $1e-2$ | – | $3e-5$ | 1.2 | 5.0 | $1e-2$ | – | $1e-4$ | 1.0 | 1.0 | $1e-2/1e-1$ | – | $3e-5$ | 1.2 | 5.0 |
| uPU | $1e-3$ | – | $3e-7$ | 1.2 | 5.0 | $1e-3$ | – | $1e-4$ | 1.0 | 1.0 | $1e-3/1e-1$ | – | $3e-7$ | 1.2 | 5.0 |
| nnPU | $1e-3$ | – | $3e-7$ | 1.2 | 5.0 | $1e-3$ | – | $1e-4$ | 1.0 | 1.0 | $1e-3/1e-1$ | – | $3e-7$ | 1.2 | 5.0 |
| BODA | $1e-3$ | – | $3e-3$ | 1.2 | 5.0 | $1e-3$ | – | $1e-4$ | 1.0 | 1.0 | $1e-3/1e-2$ | – | $3e-3$ | 1.2 | 5.0 |
| ARPL/ARPL+CS | $1e-2$ | – | $3e-5$ | 1.0 | 100.0 | – | – | – | – | – | $1e-2$ | – | $3e-5$ | 1.0 | 100.0 |
| CoLOR | $1e-3$ | $2e-2$ | $3e-7$ | 1.2 | 5.0 | $1e-3$ | $6e-2$ | $1e-4$ | 1.0 | 1.0 | $1e-3$ | $2e-2$ | $3e-7$ | 1.2 | 5.0 |

Table 5: SUN397 dataset with distribution shift due to varying proportions of subtypes of scenes/places. All the methods here use ResNet50 backbone pretrained on ImageNet1K_V1 Russakovsky et al. (2015). AUROC and AUPRC represent the performance for the novel category detection task while OSCR measures overall performance of the methods on both known and unknown classes. $\alpha$ is the mixture proportion column for the respective novel classes.

| Metric | Method | Novel Classes (natural outdoor scenes/places) | | | |
|---|---|---|---|---|---|
| | | $\alpha = 0.06 \pm 0.01$ | $\alpha = 0.06 \pm 0.01$ | $\alpha = 0.08 \pm 0.04$ | |
| | | [water, ice, snow, etc.] | [mountains, hills, desert, sky, etc.] | [forest, field, jungle, etc.] | Summary |
| AUROC | DD | $0.90 \pm 0.06$ | $0.88 \pm 0.05$ | $0.94 \pm 0.02$ | $0.91 \pm 0.05$ |
| | uPU | $0.71 \pm 0.13$ | $0.72 \pm 0.18$ | $0.84 \pm 0.08$ | $0.76 \pm 0.14$ |
| | nnPU | $0.71 \pm 0.13$ | $0.72 \pm 0.18$ | $0.84 \pm 0.08$ | $0.76 \pm 0.14$ |
| | BODA | $0.86 \pm 0.03$ | $0.90 \pm 0.01$ | $0.82 \pm 0.09$ | $0.86 \pm 0.06$ |
| | CoLOR | $\mathbf{0.98 \pm 0.02}$ | $\mathbf{0.98 \pm 0.01}$ | $\mathbf{0.98 \pm 0.02}$ | $\mathbf{0.98 \pm 0.02}$ |
| AUPRC | DD | $0.50 \pm 0.20$ | $0.40 \pm 0.23$ | $0.73 \pm 0.10$ | $0.54 \pm 0.22$ |
| | uPU | $0.11 \pm 0.04$ | $0.18 \pm 0.20$ | $0.34 \pm 0.30$ | $0.21 \pm 0.22$ |
| | nnPU | $0.11 \pm 0.04$ | $0.18 \pm 0.20$ | $0.34 \pm 0.30$ | $0.21 \pm 0.22$ |
| | BODA | $0.41 \pm 0.04$ | $0.52 \pm 0.16$ | $0.43 \pm 0.16$ | $0.45 \pm 0.11$ |
| | CoLOR | $\mathbf{0.92 \pm 0.04}$ | $\mathbf{0.93 \pm 0.05}$ | $\mathbf{0.89 \pm 0.15}$ | $\mathbf{0.91 \pm 0.09}$ |
| OSCR | DD | $0.66 \pm 0.04$ | $0.67 \pm 0.06$ | $0.70 \pm 0.03$ | $0.68 \pm 0.05$ |
| | uPU | $0.35 \pm 0.10$ | $0.35 \pm 0.08$ | $0.48 \pm 0.11$ | $0.40 \pm 0.11$ |
| | nnPU | $0.35 \pm 0.10$ | $0.35 \pm 0.08$ | $0.48 \pm 0.11$ | $0.40 \pm 0.11$ |
| | BODA | $0.56 \pm 0.10$ | $0.59 \pm 0.09$ | $0.50 \pm 0.09$ | $0.55 \pm 0.10$ |
| | CoLOR | $\mathbf{0.82 \pm 0.03}$ | $\mathbf{0.81 \pm 0.03}$ | $\mathbf{0.81 \pm 0.05}$ | $\mathbf{0.81 \pm 0.04}$ |

Table 6: SUN397 dataset with distribution shift due to varying proportions of subtypes of scenes/places. All the principled methods (DD, uPU, nnPU, BODA & CoLOR) here use pretrained CLIP ViT-L/14 backbone from Radford et al. (2021).

| Metric | Method | Novel Classes (natural outdoor scenes/places) | | | |
|---|---|---|---|---|---|
| | | $\alpha = 0.06 \pm 0.01$ | $\alpha = 0.06 \pm 0.01$ | $\alpha = 0.08 \pm 0.04$ | |
| | | [water, ice, snow, etc.] | [mountains, hills, desert, sky, etc.] | [forest, field, jungle, etc.] | Summary |
| AUROC | DD | $0.96 \pm 0.03$ | $0.96 \pm 0.03$ | $0.96 \pm 0.02$ | $0.96 \pm 0.02$ |
| | uPU | $0.95 \pm 0.03$ | $0.94 \pm 0.05$ | $0.95 \pm 0.04$ | $0.95 \pm 0.04$ |
| | nnPU | $0.95 \pm 0.03$ | $0.94 \pm 0.05$ | $0.95 \pm 0.04$ | $0.95 \pm 0.04$ |
| | BODA | $0.91 \pm 0.06$ | $0.82 \pm 0.18$ | $0.79 \pm 0.19$ | $0.84 \pm 0.15$ |
| | ARPL | $0.73 \pm 0.03$ | $0.72 \pm 0.08$ | $0.68 \pm 0.12$ | $0.71 \pm 0.08$ |
| | ZOC | $0.84 \pm 0.02$ | $0.85 \pm 0.03$ | $0.78 \pm 0.12$ | $0.82 \pm 0.07$ |
| | CoLOR | $\mathbf{0.99 \pm 0.02}$ | $\mathbf{0.99 \pm 0.01}$ | $\mathbf{0.99 \pm 0.01}$ | $\mathbf{0.99 \pm 0.01}$ |
| AUPRC | DD | $0.88 \pm 0.05$ | $0.85 \pm 0.12$ | $0.90 \pm 0.07$ | $0.87 \pm 0.08$ |
| | uPU | $0.84 \pm 0.05$ | $0.78 \pm 0.19$ | $0.85 \pm 0.10$ | $0.82 \pm 0.12$ |
| | nnPU | $0.84 \pm 0.05$ | $0.78 \pm 0.19$ | $0.85 \pm 0.10$ | $0.82 \pm 0.12$ |
| | BODA | $0.39 \pm 0.23$ | $0.37 \pm 0.39$ | $0.38 \pm 0.30$ | $0.38 \pm 0.29$ |
| | ARPL | $0.11 \pm 0.02$ | $0.11 \pm 0.04$ | $0.15 \pm 0.11$ | $0.12 \pm 0.07$ |
| | ZOC | $0.21 \pm 0.05$ | $0.23 \pm 0.04$ | $0.26 \pm 0.08$ | $0.23 \pm 0.06$ |
| | CoLOR | $\mathbf{0.95 \pm 0.04}$ | $\mathbf{0.95 \pm 0.01}$ | $\mathbf{0.96 \pm 0.03}$ | $\mathbf{0.95 \pm 0.03}$ |
| OSCR | DD | $0.93 \pm 0.02$ | $0.93 \pm 0.03$ | $0.93 \pm 0.02$ | $0.93 \pm 0.02$ |
| | uPU | $0.92 \pm 0.02$ | $0.91 \pm 0.05$ | $0.91 \pm 0.04$ | $0.92 \pm 0.03$ |
| | nnPU | $0.92 \pm 0.02$ | $0.91 \pm 0.05$ | $0.91 \pm 0.04$ | $0.92 \pm 0.03$ |
| | BODA | $0.82 \pm 0.04$ | $0.61 \pm 0.34$ | $0.73 \pm 0.14$ | $0.72 \pm 0.21$ |
| | ARPL | $0.61 \pm 0.01$ | $0.61 \pm 0.06$ | $0.58 \pm 0.08$ | $0.60 \pm 0.06$ |
| | ZOC | $0.53 \pm 0.03$ | $0.52 \pm 0.05$ | $0.47 \pm 0.08$ | $0.51 \pm 0.06$ |
| | CoLOR | $\mathbf{0.96 \pm 0.02}$ | $\mathbf{0.97 \pm 0.01}$ | $\mathbf{0.96 \pm 0.01}$ | $\mathbf{0.96 \pm 0.01}$ |

Table 7: SUN397 dataset **without** any intended distribution shift. All the principled methods (DD, uPU, nnPU, BODA & CoLOR) here use pretrained CLIP ViT-L/14 backbone from Radford et al. (2021).

| Metric | Method | Novel Classes (natural outdoor scenes/places) | | | |
| | | $\alpha = 0.06 \pm 0.00$ | $\alpha = 0.05 \pm 0.01$ | $\alpha = 0.07 \pm 0.03$ | |
| | | [water, ice, snow, etc.] | [mountains, hills, desert, sky, etc.] | [forest, field, jungle, etc.] | Summary |
| AUROC | DD | $1.00 \pm 0.00$ | $1.00 \pm 0.00$ | $1.00 \pm 0.00$ | $1.00 \pm 0.00$ |
| | uPU | $1.00 \pm 0.00$ | $1.00 \pm 0.00$ | $1.00 \pm 0.00$ | $1.00 \pm 0.00$ |
| | nnPU | $1.00 \pm 0.00$ | $1.00 \pm 0.00$ | $1.00 \pm 0.00$ | $1.00 \pm 0.00$ |
| | BODA | $0.87 \pm 0.03$ | $0.88 \pm 0.04$ | $0.90 \pm 0.05$ | $0.88 \pm 0.04$ |
| | ARPL | $0.86 \pm 0.03$ | $0.81 \pm 0.03$ | $0.84 \pm 0.05$ | $0.84 \pm 0.04$ |
| | ZOC | $0.84 \pm 0.02$ | $0.86 \pm 0.02$ | $0.77 \pm 0.12$ | $0.82 \pm 0.08$ |
| | CoLOR | $1.00 \pm 0.00$ | $1.00 \pm 0.00$ | $1.00 \pm 0.00$ | $1.00 \pm 0.00$ |
| AUPRC | DD | $1.00 \pm 0.00$ | $1.0 \pm 0.00$ | $1.00 \pm 0.00$ | $1.00 \pm 0.00$ |
| | uPU | $1.00 \pm 0.00$ | $1.00 \pm 0.00$ | $1.00 \pm 0.00$ | $1.00 \pm 0.00$ |
| | nnPU | $1.00 \pm 0.00$ | $1.00 \pm 0.00$ | $1.00 \pm 0.00$ | $1.00 \pm 0.00$ |
| | BODA | $0.18 \pm 0.04$ | $0.20 \pm 0.06$ | $0.34 \pm 0.22$ | $0.24 \pm 0.15$ |
| | ARPL | $0.20 \pm 0.04$ | $0.16 \pm 0.04$ | $0.25 \pm 0.09$ | $0.20 \pm 0.07$ |
| | ZOC | $0.19 \pm 0.02$ | $0.21 \pm 0.04$ | $0.25 \pm 0.11$ | $0.22 \pm 0.07$ |
| | CoLOR | $0.99 \pm 0.01$ | $0.99 \pm 0.01$ | $0.99 \pm 0.01$ | $0.99 \pm 0.01$ |
| OSCR | DD | $0.99 \pm 0.00$ | $0.99 \pm 0.00$ | $0.99 \pm 0.00$ | $0.99 \pm 0.00$ |
| | uPU | $0.99 \pm 0.00$ | $0.99 \pm 0.00$ | $0.99 \pm 0.00$ | $0.99 \pm 0.00$ |
| | nnPU | $0.99 \pm 0.00$ | $0.99 \pm 0.00$ | $0.99 \pm 0.00$ | $0.99 \pm 0.00$ |
| | BODA | $0.94 \pm 0.01$ | $0.94 \pm 0.01$ | $0.92 \pm 0.02$ | $0.93 \pm 0.02$ |
| | ARPL | $0.82 \pm 0.02$ | $0.78 \pm 0.03$ | $0.80 \pm 0.05$ | $0.80 \pm 0.04$ |
| | ZOC | $0.50 \pm 0.01$ | $0.51 \pm 0.02$ | $0.45 \pm 0.07$ | $0.49 \pm 0.05$ |
| | CoLOR | $0.98 \pm 0.00$ | $0.98 \pm 0.00$ | $0.98 \pm 0.00$ | $0.98 \pm 0.00$ |

Table 8: CIFAR100 dataset with background shift due to varying proportions of subtypes. (AUROC) and (AUPRC) represent the performance for the task of novel class detection while Open-Set Classification Rate (OSCR) measures overall performance of the methods on both known and unknown classes. $\alpha$ is in the range of 0.05 to 0.10. *Note that all the adaptive methods use ResNet18 and are trained from scratch.*

| Metric | Method | Novel Classes ($\alpha = 0.07 \pm 0.02$) | | | | | |
| | | Baby | Man | Butterfly | Rocket | Streetcar | Summary |
| AUROC | DD | $0.76 \pm 0.03$ | $0.61 \pm 0.13$ | $0.64 \pm 0.10$ | $0.78 \pm 0.09$ | $0.72 \pm 0.07$ | $0.70 \pm 0.11$ |
| | uPU | $0.71 \pm 0.07$ | $0.62 \pm 0.09$ | $0.56 \pm 0.13$ | $0.74 \pm 0.06$ | $0.73 \pm 0.05$ | $0.67 \pm 0.10$ |
| | nnPU | $0.69 \pm 0.09$ | $0.58 \pm 0.10$ | $0.60 \pm 0.09$ | $0.74 \pm 0.06$ | $0.72 \pm 0.16$ | $0.67 \pm 0.12$ |
| | BODA | $0.57 \pm 0.07$ | $0.54 \pm 0.05$ | $0.59 \pm 0.03$ | $0.54 \pm 0.07$ | $0.61 \pm 0.05$ | $0.57 \pm 0.06$ |
| | ARPL+CS | $0.80 \pm 0.02$ | $0.84 \pm 0.02$ | $0.71 \pm 0.02$ | $0.76 \pm 0.05$ | $0.77 \pm 0.05$ | $0.78 \pm 0.05$ |
| | ZOC | $0.97 \pm 0.01$ | $0.98 \pm 0.01$ | $0.82 \pm 0.03$ | $0.98 \pm 0.01$ | $1.00 \pm 0.00$ | $0.95 \pm 0.07$ |
| | CoLOR | $0.77 \pm 0.06$ | $0.70 \pm 0.12$ | $0.68 \pm 0.04$ | $0.83 \pm 0.04$ | $0.85 \pm 0.03$ | $0.77 \pm 0.09$ |
| AUPRC | DD | $0.25 \pm 0.03$ | $0.15 \pm 0.10$ | $0.15 \pm 0.09$ | $0.37 \pm 0.12$ | $0.27 \pm 0.17$ | $0.24 \pm 0.13$ |
| | uPU | $0.22 \pm 0.18$ | $0.11 \pm 0.03$ | $0.11 \pm 0.05$ | $0.28 \pm 0.14$ | $0.18 \pm 0.10$ | $0.18 \pm 0.12$ |
| | nnPU | $0.21 \pm 0.16$ | $0.09 \pm 0.03$ | $0.11 \pm 0.04$ | $0.32 \pm 0.09$ | $0.27 \pm 0.25$ | $0.20 \pm 0.16$ |
| | BODA | $0.10 \pm 0.05$ | $0.08 \pm 0.02$ | $0.09 \pm 0.02$ | $0.08 \pm 0.02$ | $0.09 \pm 0.03$ | $0.09 \pm 0.03$ |
| | ARPL+CS | $0.25 \pm 0.07$ | $0.29 \pm 0.08$ | $0.16 \pm 0.04$ | $0.17 \pm 0.07$ | $0.18 \pm 0.08$ | $0.21 \pm 0.08$ |
| | ZOC | $0.89 \pm 0.04$ | $0.76 \pm 0.05$ | $0.32 \pm 0.07$ | $0.77 \pm 0.05$ | $0.96 \pm 0.02$ | $0.74 \pm 0.23$ |
| | CoLOR | $0.35 \pm 0.12$ | $0.20 \pm 0.11$ | $0.21 \pm 0.06$ | $0.43 \pm 0.03$ | $0.44 \pm 0.14$ | $0.33 \pm 0.14$ |
| OSCR | DD | $0.55 \pm 0.04$ | $0.44 \pm 0.12$ | $0.44 \pm 0.11$ | $0.57 \pm 0.10$ | $0.52 \pm 0.07$ | $0.51 \pm 0.10$ |
| | uPU | $0.51 \pm 0.08$ | $0.45 \pm 0.08$ | $0.42 \pm 0.12$ | $0.53 \pm 0.05$ | $0.53 \pm 0.05$ | $0.49 \pm 0.09$ |
| | nnPU | $0.50 \pm 0.09$ | $0.41 \pm 0.07$ | $0.44 \pm 0.07$ | $0.54 \pm 0.08$ | $0.52 \pm 0.16$ | $0.48 \pm 0.10$ |
| | BODA | $0.61 \pm 0.04$ | $0.61 \pm 0.04$ | $0.61 \pm 0.02$ | $0.57 \pm 0.05$ | $0.62 \pm 0.04$ | $0.60 \pm 0.04$ |
| | ARPL+CS | $0.68 \pm 0.01$ | $0.70 \pm 0.02$ | $0.61 \pm 0.02$ | $0.65 \pm 0.02$ | $0.66 \pm 0.04$ | $0.66 \pm 0.04$ |
| | ZOC | $0.82 \pm 0.02$ | $0.83 \pm 0.02$ | $0.70 \pm 0.02$ | $0.82 \pm 0.02$ | $0.82 \pm 0.01$ | $0.84 \pm 0.01$ |
| | CoLOR | $0.58 \pm 0.06$ | $0.54 \pm 0.12$ | $0.52 \pm 0.05$ | $0.64 \pm 0.05$ | $0.65 \pm 0.02$ | $0.59 \pm 0.08$ |

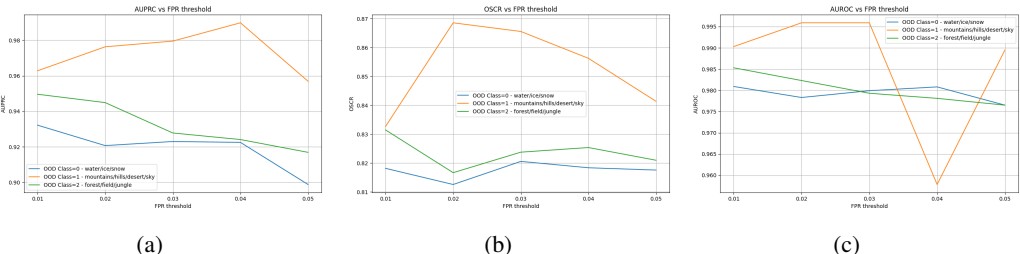

(a)      (b)      (c)

Figure 3: Effects of varying the FPR threshold on CoLOR method performance on SUN397 dataset.

Table 9: CIFAR100 dataset without background shift. (AUROC) and (AUPRC) represent the performance for the task of novel class detection while Open-Set Classification Rate (OSCR) measures overall performance of the methods on both known and unknown classes. $\alpha$ is set to 0.05. *Note that all the adaptive methods use ResNet18 and are trained from scratch.*

| Metric | Method | Novel Classes ($\alpha= 0.05 \pm 0.00$) | | | | | Summary |
|---|---|---|---|---|---|---|---|
| | | Baby | Man | Butterfly | Rocket | Streetcar | |
| AUROC | DD | $0.82 \pm 0.04$ | $0.71 \pm 0.09$ | $0.71 \pm 0.11$ | $0.84 \pm 0.09$ | $0.72 \pm 0.15$ | $0.76 \pm 0.11$ |
| | uPU | $0.75 \pm 0.10$ | $0.67 \pm 0.08$ | $0.64 \pm 0.12$ | $0.84 \pm 0.06$ | $0.74 \pm 0.11$ | $0.73 \pm 0.11$ |
| | nnPU | $0.75 \pm 0.05$ | $0.62 \pm 0.05$ | $0.66 \pm 0.07$ | $0.82 \pm 0.04$ | $0.79 \pm 0.08$ | $0.73 \pm 0.10$ |
| | BODA | $0.59 \pm 0.0$ | $0.58 \pm 0.02$ | $0.59 \pm 0.05$ | $0.59 \pm 0.02$ | $0.62 \pm 0.06$ | $0.60 \pm 0.04$ |
| | ARPL+CS | $0.79 \pm 0.02$ | $0.83 \pm 0.01$ | $0.72 \pm 0.03$ | $0.74 \pm 0.02$ | $0.78 \pm 0.00$ | $0.77 \pm 0.04$ |
| | ZOC | $\mathbf{0.97 \pm 0.01}$ | $\mathbf{0.98 \pm 0.01}$ | $\mathbf{0.82 \pm 0.02}$ | $\mathbf{0.98 \pm 0.00}$ | $\mathbf{1.00 \pm 0.00}$ | $\mathbf{0.95 \pm 0.07}$ |
| | CoLOR | $0.82 \pm 0.05$ | $0.76 \pm 0.04$ | $0.76 \pm 0.05$ | $0.82 \pm 0.03$ | $0.82 \pm 0.05$ | $0.80 \pm 0.05$ |
| AUPRC | DD | $0.39 \pm 0.10$ | $0.15 \pm 0.06$ | $0.20 \pm 0.09$ | $0.48 \pm 0.23$ | $0.25 \pm 0.16$ | $0.29 \pm 0.18$ |
| | uPU | $0.20 \pm 0.11$ | $0.13 \pm 0.06$ | $0.11 \pm 0.05$ | $0.45 \pm 0.12$ | $0.21 \pm 0.13$ | $0.22 \pm 0.15$ |
| | nnPU | $0.18 \pm 0.08$ | $0.08 \pm 0.02$ | $0.10 \pm 0.05$ | $0.39 \pm 0.12$ | $0.27 \pm 0.14$ | $0.21 \pm 0.14$ |
| | BODA | $0.07 \pm 0.01$ | $0.06 \pm 0.00$ | $0.06 \pm 0.01$ | $0.07 \pm 0.01$ | $0.07 \pm 0.01$ | $0.07 \pm 0.01$ |
| | ARPL+CS | $0.17 \pm 0.02$ | $0.20 \pm 0.02$ | $0.13 \pm 0.03$ | $0.12 \pm 0.03$ | $0.13 \pm 0.01$ | $0.15 \pm 0.04$ |
| | ZOC | $\mathbf{0.87 \pm 0.03}$ | $\mathbf{0.68 \pm 0.04}$ | $\mathbf{0.25 \pm 0.04}$ | $\mathbf{0.70 \pm 0.05}$ | $\mathbf{0.94 \pm 0.02}$ | $\mathbf{0.69 \pm 0.25}$ |
| | CoLOR | $0.35 \pm 0.12$ | $0.17 \pm 0.05$ | $0.21 \pm 0.07$ | $0.39 \pm 0.05$ | $0.33 \pm 0.11$ | $0.29 \pm 0.12$ |
| OSCR | DD | $0.63 \pm 0.05$ | $0.51 \pm 0.15$ | $0.52 \pm 0.17$ | $0.62 \pm 0.17$ | $0.53 \pm 0.18$ | $0.56 \pm 0.15$ |
| | uPU | $0.54 \pm 0.12$ | $0.49 \pm 0.08$ | $0.45 \pm 0.11$ | $0.63 \pm 0.06$ | $0.53 \pm 0.13$ | $0.53 \pm 0.12$ |
| | nnPU | $0.55 \pm 0.06$ | $0.44 \pm 0.06$ | $0.45 \pm 0.07$ | $0.58 \pm 0.06$ | $0.59 \pm 0.10$ | $0.52 \pm 0.09$ |
| | BODA | $0.62 \pm 0.04$ | $0.63 \pm 0.03$ | $0.61 \pm 0.02$ | $0.56 \pm 0.03$ | $0.64 \pm 0.01$ | $0.61 \pm 0.04$ |
| | ARPL+CS | $0.68 \pm 0.01$ | $0.71 \pm 0.01$ | $0.63 \pm 0.02$ | $0.64 \pm 0.02$ | $0.68 \pm 0.01$ | $0.67 \pm 0.03$ |
| | ZOC | $\mathbf{0.81 \pm 0.01}$ | $\mathbf{0.81 \pm 0.00}$ | $\mathbf{0.70 \pm 0.02}$ | $\mathbf{0.81 \pm 0.00}$ | $\mathbf{0.82 \pm 0.01}$ | $\mathbf{0.79 \pm 0.05}$ |
| | CoLOR | $0.65 \pm 0.02$ | $0.60 \pm 0.03$ | $0.59 \pm 0.05$ | $0.63 \pm 0.04$ | $0.63 \pm 0.06$ | $0.62 \pm 0.04$ |

Table 10: Amazon Reviews dataset with sentiment based distribution shift. AUROC and AUPRC represent the performance for the task of novel class detection while OSCR measures overall performance of the methods on both known and unknown classes. $\alpha$ is the mixture proportion column for the respective novel classes.

| Metric | Method | Novel Classes | | | | | | |
|---|---|---|---|---|---|---|---|---|
| | | $\alpha = 0.32$ | $\alpha = 0.11$ | $\alpha = 0.07$ | $\alpha = 0.07$ | $\alpha = 0.22$ | $\alpha = 0.18$ | |
| | | Musical Instruments | Digital Music | Software | Luxury Beauty | Industrial & Scientific | Prime Pantry | Summary |
| AUROC | DD | $0.75 \pm 0.05$ | $0.81 \pm 0.04$ | $0.80 \pm 0.04$ | $0.73 \pm 0.04$ | $0.58 \pm 0.04$ | $0.63 \pm 0.06$ | $0.72 \pm 0.09$ |
| | uPU | $0.79 \pm 0.04$ | $0.85 \pm 0.06$ | $0.79 \pm 0.03$ | $0.76 \pm 0.03$ | $0.67 \pm 0.04$ | $0.67 \pm 0.09$ | $0.76 \pm 0.08$ |
| | nnPU | $0.79 \pm 0.04$ | $0.85 \pm 0.06$ | $0.79 \pm 0.03$ | $0.76 \pm 0.03$ | $0.67 \pm 0.04$ | $0.67 \pm 0.09$ | $0.76 \pm 0.08$ |
| | BODA | $0.76 \pm 0.03$ | $0.61 \pm 0.13$ | $0.56 \pm 0.06$ | $0.67 \pm 0.07$ | $0.72 \pm 0.04$ | $0.65 \pm 0.09$ | $0.66 \pm 0.10$ |
| | CoLOR | $\mathbf{0.82 \pm 0.03}$ | $\mathbf{0.87 \pm 0.04}$ | $\mathbf{0.84 \pm 0.06}$ | $\mathbf{0.77 \pm 0.04}$ | $\mathbf{0.68 \pm 0.09}$ | $\mathbf{0.72 \pm 0.11}$ | $\mathbf{0.79 \pm 0.09}$ |
| AUPRC | DD | $0.68 \pm 0.06$ | $0.62 \pm 0.08$ | $0.32 \pm 0.12$ | $0.27 \pm 0.09$ | $0.33 \pm 0.05$ | $0.38 \pm 0.07$ | $0.43 \pm 0.18$ |
| | uPU | $0.73 \pm 0.04$ | $0.68 \pm 0.10$ | $0.39 \pm 0.07$ | $0.37 \pm 0.10$ | $0.43 \pm 0.07$ | $0.44 \pm 0.14$ | $0.51 \pm 0.17$ |
| | nnPU | $0.73 \pm 0.04$ | $0.68 \pm 0.10$ | $0.39 \pm 0.07$ | $0.37 \pm 0.10$ | $0.43 \pm 0.07$ | $0.44 \pm 0.14$ | $0.51 \pm 0.17$ |
| | BODA | $0.67 \pm 0.07$ | $0.19 \pm 0.11$ | $0.07 \pm 0.01$ | $0.14 \pm 0.04$ | $0.38 \pm 0.06$ | $0.25 \pm 0.07$ | $0.28 \pm 0.21$ |
| | CoLOR | $\mathbf{0.76 \pm 0.04}$ | $\mathbf{0.69 \pm 0.07}$ | $\mathbf{0.49 \pm 0.12}$ | $\mathbf{0.35 \pm 0.14}$ | $\mathbf{0.44 \pm 0.12}$ | $\mathbf{0.50 \pm 0.15}$ | $\mathbf{0.54 \pm 0.18}$ |
| OSCR | DD | $0.54 \pm 0.04$ | $0.54 \pm 0.05$ | $0.53 \pm 0.05$ | $0.47 \pm 0.02$ | $0.43 \pm 0.03$ | $0.46 \pm 0.04$ | $0.50 \pm 0.06$ |
| | uPU | $0.59 \pm 0.03$ | $0.59 \pm 0.06$ | $0.53 \pm 0.04$ | $0.49 \pm 0.02$ | $0.52 \pm 0.03$ | $0.51 \pm 0.03$ | $0.54 \pm 0.05$ |
| | nnPU | $0.59 \pm 0.03$ | $0.59 \pm 0.06$ | $0.53 \pm 0.04$ | $0.49 \pm 0.02$ | $0.52 \pm 0.03$ | $0.51 \pm 0.03$ | $0.54 \pm 0.05$ |
| | BODA | $0.50 \pm 0.01$ | $0.53 \pm 0.05$ | $\mathbf{0.56 \pm 0.04}$ | $\mathbf{0.51 \pm 0.03}$ | $\mathbf{0.62 \pm 0.01}$ | $\mathbf{0.62 \pm 0.06}$ | $\mathbf{0.56 \pm 0.06}$ |
| | CoLOR | $\mathbf{0.61 \pm 0.04}$ | $\mathbf{0.59 \pm 0.05}$ | $0.56 \pm 0.07$ | $0.50 \pm 0.05$ | $0.56 \pm 0.03$ | $0.55 \pm 0.07$ | $\mathbf{0.56 \pm 0.06}$ |

Table 11: AUPRC scores with & w/o joint learning for novelty detection on CIFAR100 dataset with distribution shift.

| Novel Class | $\alpha$ | Absolute AUPRC | | | | | | | |
|---|---|---|---|---|---|---|---|---|---|
| | | DD w/o joint learning | DD | uPU w/o joint learning | uPU | nnPU w/o joint learning | nnPU | CoLoR w/o joint learning | CoLOR |
| Baby | $0.23 \pm 0.05$ | $0.69 \pm 0.10$ | $0.77 \pm 0.04$ | $0.63 \pm 0.10$ | $0.64 \pm 0.10$ | $0.61 \pm 0.08$ | $0.61 \pm 0.11$ | $0.65 \pm 0.13$ | $\mathbf{0.80 \pm 0.08}$ |
| Tulip | $0.29 \pm 0.12$ | $0.41 \pm 0.14$ | $0.39 \pm 0.14$ | $0.27 \pm 0.11$ | $0.25 \pm 0.04$ | $0.26 \pm 0.12$ | $0.29 \pm 0.10$ | $0.50 \pm 0.16$ | $\mathbf{0.53 \pm 0.16}$ |
| Crocodile | $0.23 \pm 0.05$ | $0.58 \pm 0.08$ | $0.64 \pm 0.04$ | $0.47 \pm 0.09$ | $0.44 \pm 0.12$ | $0.47 \pm 0.11$ | $0.42 \pm 0.11$ | $0.61 \pm 0.15$ | $\mathbf{0.68 \pm 0.10}$ |
| Dolphin | $0.29 \pm 0.12$ | $\mathbf{0.65 \pm 0.12}$ | $\mathbf{0.65 \pm 0.12}$ | $0.50 \pm 0.06$ | $0.49 \pm 0.08$ | $0.50 \pm 0.08$ | $0.45 \pm 0.11$ | $\mathbf{0.65 \pm 0.12}$ | $0.64 \pm 0.19$ |
| Man | $0.23 \pm 0.05$ | $0.59 \pm 0.19$ | $0.64 \pm 0.13$ | $0.31 \pm 0.04$ | $0.41 \pm 0.14$ | $0.28 \pm 0.12$ | $0.46 \pm 0.13$ | $0.68 \pm 0.09$ | $\mathbf{0.79 \pm 0.07}$ |

Table 12: AUPRC scores with & w/o joint learning for novelty detection on Amazon Reviews dataset with distribution shift.

| Novel Class | $\alpha$ | Absolute AUPRC | | | | | | | |
|---|---|---|---|---|---|---|---|---|---|
| | | DD w/o joint learning | DD | uPU w/o joint learning | uPU | nnPU w/o joint learning | nnPU | CoLoR w/o joint learning | CoLOR |
| Musical Instruments | 0.16 | $0.37 \pm 0.09$ | $0.40 \pm 0.1$ | $0.34 \pm 0.09$ | $0.45 \pm 0.13$ | $0.34 \pm 0.09$ | $0.45 \pm 0.13$ | $0.48 \pm 0.11$ | $\mathbf{0.55 \pm 0.13}$ |
| Digital Music | 0.05 | $0.28 \pm 0.07$ | $0.29 \pm 0.09$ | $0.23 \pm 0.10$ | $0.24 \pm 0.12$ | $0.23 \pm 0.10$ | $0.24 \pm 0.12$ | $0.33 \pm 0.13$ | $\mathbf{0.35 \pm 0.15}$ |
| Software | 0.06 | $0.36 \pm 0.05$ | $0.37 \pm 0.11$ | $0.31 \pm 0.13$ | $0.36 \pm 0.08$ | $0.31 \pm 0.13$ | $0.36 \pm 0.08$ | $0.48 \pm 0.11$ | $\mathbf{0.49 \pm 0.12}$ |
| Luxury Beauty | 0.07 | $0.27 \pm 0.15$ | $0.25 \pm 0.08$ | $0.28 \pm 0.12$ | $0.35 \pm 0.11$ | $0.28 \pm 0.12$ | $0.35 \pm 0.11$ | $0.35 \pm 0.14$ | $\mathbf{0.36 \pm 0.13}$ |
| Industrial & Scientific | 0.12 | $0.13 \pm 0.01$ | $0.14 \pm 0.03$ | $0.11 \pm 0.02$ | $0.16 \pm 0.05$ | $0.11 \pm 0.02$ | $0.16 \pm 0.05$ | $0.17 \pm 0.02$ | $\mathbf{0.21 \pm 0.08}$ |

Table 13: MSP/Entropy based OOD detection on SUN397 with low $\alpha = 0.05 \pm 0.01$ on three types of novel categories

|  | [water, ice, snow, etc.] | | [mountains, hills, desert, sky, etc.] | | Class [forest, field, jungle, etc.] | |
|---|---|---|---|---|---|---|
|  | AUROC | AUPRC | AUROC | AUPRC | AUROC | AUPRC |
| With Shift | | | | | | |
| MSP | $0.57 \pm 0.05$ | $0.05 \pm 0.01$ | $0.58 \pm 0.06$ | $0.05 \pm 0.01$ | $0.56 \pm 0.05$ | $0.07 \pm 0.03$ |
| Entropy | $0.58 \pm 0.07$ | $0.08 \pm 0.04$ | $0.58 \pm 0.08$ | $0.07 \pm 0.03$ | $0.56 \pm 0.07$ | $0.10 \pm 0.06$ |
| Without Shift | | | | | | |
| MSP | $0.59 \pm 0.05$ | $0.04 \pm 0.00$ | $0.59 \pm 0.05$ | $0.05 \pm 0.01$ | $0.58 \pm 0.07$ | $0.06 \pm 0.03$ |
| Entropy | $0.59 \pm 0.07$ | $0.08 \pm 0.02$ | $0.59 \pm 0.08$ | $0.08 \pm 0.04$ | $0.59 \pm 0.10$ | $0.09 \pm 0.03$ |

Table 14: MSP/Entropy based OOD detection on SUN397 with high $\alpha = 0.5 \pm 0.10$ on three types of novel categories

|  | Class 0 | | Class 1 | | Class 2 | |
|---|---|---|---|---|---|---|
|  | AUROC | AUPRC | AUROC | AUPRC | AUROC | AUPRC |
| With Shift | | | | | | |
| MSP | $0.52 \pm 0.06$ | $0.55 \pm 0.05$ | $0.61 \pm 0.06$ | $0.28 \pm 0.04$ | $0.60 \pm 0.08$ | $0.50 \pm 0.09$ |
| Entropy | $0.52 \pm 0.06$ | $0.58 \pm 0.05$ | $0.63 \pm 0.06$ | $0.45 \pm 0.07$ | $0.62 \pm 0.09$ | $0.67 \pm 0.04$ |
| Without Shift | | | | | | |
| MSP | $0.59 \pm 0.06$ | $0.47 \pm 0.04$ | $0.61 \pm 0.08$ | $0.27 \pm 0.04$ | $0.58 \pm 0.08$ | $0.50 \pm 0.05$ |
| Entropy | $0.59 \pm 0.08$ | $0.59 \pm 0.07$ | $0.63 \pm 0.10$ | $0.41 \pm 0.10$ | $0.59 \pm 0.10$ | $0.63 \pm 0.09$ |

Table 15: Effect of background shift on Top-1 accuracies of remaining baselines. We do not expect any performance reduction of ZOC as it is a zero-shot method that does not utilize source data at all, rendering any distribution shift between $P_{\mathcal{S}}$ and $P_{\mathcal{T}}$ irrelevant. Such models are influenced only by target datasets that drift from their pretraining datasets.

| Methods | w/ DS | w/o DS |
|---|---|---|
| BODA (ViT-L/14) | $0.75 \pm 0.13$ | $0.94 \pm 0.00$ |
| ARPL (custom default) | $0.75 \pm 0.03$ | $0.91 \pm 0.01$ |
| ZOC (custom default) | $0.62 \pm 0.05$ | $0.59 \pm 0.00$ |
| ANNA (custom default) | $0.65 \pm 0.04$ | $0.87 \pm 0.01$ |

Table 16: CoLOR performance for different target recall constraints for the SUN397 dataset with novel identity as outdoor scenes from the group [water, ice, snow, etc.] and novel class size $\alpha = 0.06 \pm 0.01$ using ResNet50 model. Note that out of $\|\boldsymbol{\alpha}\|$ novelty detection heads each corresponding to a candidate value $\hat{\alpha} \in \boldsymbol{\alpha}$, we select the head achieving highest recall in the validation set ($\text{argmax}_{w_{\hat{\alpha}}^{\alpha}:\alpha \in \boldsymbol{\alpha}, \hat{\beta}(w_{\hat{\alpha}}^{\alpha}) < \beta} \hat{\alpha}(w_{\hat{\alpha}}^{\alpha})$). Consequently, the reported AUROC, AUPRC and OSCR performance correspond to that selected head. Scores of the selected novelty detection head for each seed are highlighted in bold below whereas the highest scores amongst all the novelty detection heads for each seed have been underlined. These heads having scores underlined were not selected besides having highest scores for a metric, because they simply did not satisfy the selection criteria, i.e. their validation recall values were not the highest for the corresponding seed among all the novelty detection heads.

| Target Recall | Validation Recall across 5 seeds | | | | | AUROC across 5 seeds | | | | | AUPRC across 5 seeds | | | | | OSCR across 5 seeds | | | | |
|---|---|---|---|---|---|---|---|---|---|---|---|---|---|---|---|---|---|---|---|---|
| $\hat{\alpha}$ | 0 | 8 | 1057 | 103 | 573 | 0 | 8 | 1057 | 103 | 573 | 0 | 8 | 1057 | 103 | 573 | 0 | 8 | 1057 | 103 | 573 |
| 0.02 | **0.18** | 0.06 | 0.10 | **0.08** | 0.08 | **0.98** | 0.97 | 0.95 | **0.99** | 0.94 | **0.93** | 0.87 | 0.87 | **0.94** | 0.43 | **0.82** | 0.82 | 0.75 | **0.84** | 0.67 |
| 0.05 | 0.18 | 0.06 | 0.00 | 0.08 | 0.15 | 0.99 | 0.96 | 0.48 | 1.00 | 0.99 | 0.94 | 0.82 | 0.05 | 0.95 | 0.95 | 0.83 | 0.81 | 0.25 | 0.85 | 0.78 |
| 0.10 | 0.17 | 0.02 | **0.14** | 0.06 | 0.09 | 0.98 | 0.82 | **0.95** | 1.00 | 0.93 | 0.93 | 0.13 | **0.85** | 0.96 | 0.38 | 0.81 | 0.66 | **0.76** | 0.85 | 0.65 |
| 0.15 | 0.16 | 0.10 | 0.01 | 0.06 | 0.04 | 0.99 | 0.95 | 0.42 | 0.99 | 0.95 | 0.95 | 0.80 | 0.05 | 0.92 | 0.64 | 0.84 | 0.80 | 0.23 | 0.84 | 0.65 |
| 0.20 | 0.12 | **0.12** | 0.14 | 0.05 | 0.03 | 0.97 | **0.97** | 0.95 | 0.99 | 0.93 | 0.79 | **0.90** | 0.86 | 0.94 | 0.42 | 0.76 | **0.85** | 0.78 | 0.84 | 0.64 |
| 0.25 | 0.07 | 0.00 | 0.07 | 0.03 | **0.18** | 0.99 | 0.68 | 0.47 | 0.98 | 0.99 | 0.83 | 0.08 | 0.05 | 0.80 | **0.97** | 0.77 | 0.52 | 0.27 | 0.81 | **0.82** |
| 0.30 | 0.12 | 0.01 | 0.00 | 0.07 | 0.00 | 0.99 | 0.81 | 0.44 | 1.00 | 0.86 | 0.92 | 0.13 | 0.05 | 0.95 | 0.29 | 0.77 | 0.65 | 0.25 | 0.85 | 0.61 |
| 0.35 | 0.11 | 0.04 | 0.02 | 0.01 | 0.04 | 0.97 | 0.81 | 0.47 | 0.84 | 0.83 | 0.55 | 0.13 | 0.05 | 0.21 | 0.25 | 0.74 | 0.65 | 0.26 | 0.69 | 0.59 |
| 0.40 | 0.06 | 0.00 | 0.09 | 0.01 | 0.00 | 0.94 | 0.84 | 0.43 | 0.87 | 0.78 | 0.41 | 0.16 | 0.05 | 0.25 | 0.18 | 0.73 | 0.69 | 0.23 | 0.71 | 0.57 |
| 0.45 | 0.04 | 0.00 | 0.00 | 0.07 | 0.00 | 0.99 | 0.74 | 0.54 | 0.97 | 0.79 | 0.89 | 0.11 | 0.06 | 0.82 | 0.22 | 0.77 | 0.58 | 0.34 | 0.82 | 0.57 |

