# OpenReview forum: "Open-Set Domain Adaptation Under Background Distribution Shift: Challenges and A Provably Efficient Solution"
_ICLR.cc/2025/Conference — Submitted to ICLR 2025_

### Official Review · Reviewer_Q1zn · 2024-10-28

**Soundness:** 3
**Presentation:** 2
**Contribution:** 3
**Rating:** 5
**Confidence:** 2

**Summary:**

This paper investigates Open-Set Domain Adaptation (OSDA) under a background distribution shift. In OSDA, the objective is to classify known categories while detecting novel, previously unobserved classes in the target domain. This study addresses background shifts, where the distribution of known classes in source and target domains differs. The authors present a scalable method named CoLOR (Constrained Learning for Open-Set Recognition) to robustly classify known classes and detect novel ones under this shift. They demonstrate that CoLOR outperforms existing baselines by jointly learning representations for both classification and novelty detection.

**Strengths:**

1. This paper addresses the overlooked issue of background distribution shifts in OSDA, relevant for dynamic applications like medical imaging and autonomous systems.

2. Extensive experiments across diverse datasets validate CoLOR's performance and robustness against strong baselines.

3. The paper is well-organized, with clear explanations, detailed methodology, and effective visual aids for complex concepts.

**Weaknesses:**

1. The separability assumption (Assumption 1) is central to CoLOR’s effectiveness. Could you discuss scenarios where this assumption might not hold in practical applications? How would CoLOR perform if the separability or background shift assumptions were relaxed or violated?

2. The writing quality could be enhanced by ensuring consistency in formatting, particularly in the Related Works section, where the subheading styles appear inconsistent. Standardizing these subheading formats would improve the paper’s overall readability and professionalism.

**Questions:**

Please see Weaknesses above.

---

> ### Author Response · Authors · 2024-11-27
>
> We thank the reviewer for their feedback and time taken to review the paper. We are glad to see that the reviewer finds the critical issue of background shift useful and relevant to real word applications. We further address concerns of the reviewer below:
>
> > The separability assumption (Assumption 1) is central to CoLOR’s effectiveness. Could you discuss scenarios where this assumption might not hold in practical applications? How would CoLOR perform if the separability or background shift assumptions were relaxed or violated?
>
> We restricted our formalism to the separability assumption since it is intuitive and practical. The conditions of Theorem 4.3 in [1] are more general, and handle some non-separable cases. Roughly, their condition is summarized as rare events in the source distribution not becoming ubiquitous in the target distribution. We refrained from these subtleties in our presentation, to keep the discussion simple but include a comment noting the appendix section A.2.2 of our revision , stating that the method is robust to certain violations of separability.
>
> Generally, separability is a reasonable assumption in many cases such as detecting novel visual patterns. It may not hold in cases like detecting a population with a new condition (e.g. a previously unknown medical condition), based on observable features such as vitals. These new conditions might not be pronounced in a perfectly distinct way in the observable features, but we can hope that maybe they manifest in some statistically significant manner that we can (imperfectly) detect. Our benchmarks include an example of a non-separable task, which is Amazon reviews, where classes correspond to product categories, but it can be difficult to tell from the product review’s text which product category it was written on (e.g. a review such as “great product.” does not include the necessary information). In this dataset, we still see that CoLOR performs well w.r.t baselines, which is an encouraging sign for its robustness to violations of separability.
>
> > The writing quality could be enhanced by ensuring consistency in formatting, particularly in the Related Works section, where the subheading styles appear inconsistent. Standardizing these subheading formats would improve the paper’s overall readability and professionalism.
>
> We thank the reviewer for their suggestion. We have made these changes in our latest rebuttal revision including the standardisation of subheading format. We have also improved the Introduction and Related Background section to be more concise based on the feedback from other reviewers. We have further added a new baseline ANNA [2] (designed for vision modality only) as suggested by reviewer mQDK in Table 3b.
>
> References:
>
> [1] Yoav Wald and Suchi Saria. Birds of an odd feather: guaranteed out-of-distribution (ood) novel category detection. In Uncertainty in Artificial Intelligence, pp. 2179–2191. PMLR, 2023.
>
> [2] Shuo Wen and Maria Brbic. Cross-domain open-world discovery. In Ruslan Salakhutdinov, Zico Kolter, Katherine Heller, Adrian Weller, Nuria Oliver, Jonathan Scarlett, and Felix Berkenkamp (eds.), Proceedings of the 41st International Conference on Machine Learning, volume 235 of Proceedings of Machine Learning Research, pp. 52744–52761. PMLR, 21–27 Jul 2024. URL https://proceedings.mlr.press/v235/wen24b.html.

---

> ### Author Response · Authors · 2024-11-30
> **Following up**
>
> Dear Reviewer,
>
> We thank you again for your suggestions. With the discussion period deadline approaching, we wish to know if our responses were able to adequately address your concerns or questions. If there are any further questions or concerns, we would happy to respond up until the deadline.
>
> Thank you.

---

> > ### Comment · Reviewer_Q1zn · 2024-12-01
> >
> > Thank you for your thoughtful response and for addressing my comments. I appreciate the clarification regarding the separability assumption and its practical relevance, as well as your efforts to improve the writing quality and section formatting. While your explanation provides valuable insight, I believe further discussion on handling scenarios where the separability assumption is violated would enhance the robustness of the method. I will maintain my score and look forward to seeing how this work develops further.

---

### Official Review · Reviewer_ZrBg · 2024-11-01

**Soundness:** 3
**Presentation:** 2
**Contribution:** 2
**Rating:** 5
**Confidence:** 3

**Summary:**

This work sets out to address the open set domain adaptation (OSDA) problem under so-called background shifts. This problem amounts to solving the domain adaptation problem while also identifying datapoints which are novel classes which were not seen in the source domain. Domain adaptation is the problem of finding a model which performs well in a target domain when it has been trained on labeled data from some source domain and unlabeled data from the target domain. Background shifts are defined as any distribution shift between source and target domains which retains overlapping support between source and target domain distributions with the non-novel label set.

The authors show that strong positivity and overlapping support is not a sufficient condition to ensure better than random performance.
The work proposes a model partly based on previous work in OOD detection. The work proposes to use this method to detect the novel classes while at the same time learning to do the classification. The architecture of the model has a shared base structure which is used for both the detection and classification heads.

The model is then compared on image and text dataset benchmark against several baselines. It performs favourably on these tasks compared to baselines, especially when the ratio of novel class datapoints is low.

**Strengths:**

- The setting of OSDA is both an important and challenging area

- The paper uses an OOD method in combination with learning a classifier to do OSDA which is a reasonable approach which seems to work well

- Method performs well on chosen datasets compared to baselines

**Weaknesses:**

- The introduction and related work together makes up ~3 pages which is a substantial part of the paper.

- It seems rather inefficient to solve the problem for a large number of possible values of $\alpha$. How does this compare to the other baseline methods?


- The experimental details are somewhat unclear. Some things are not clearly stated.
    * How many repetitions of experiments were done?
    * How is the separability of the novel classes ensured?

- Why does the amount of 'wins' not tally up to the total number of repetitions? I assume that this is what the Wins column in table 2 means, since I cannot find it explained in the text. Does this mean that the models tie in performance in some runs?

- Not clear to me how the choice of $\beta$ threshold impacts the results.

Typos and other comments:

- Unclear what the bolding in table 3b means, it almost seems random.

- Figure 2 is small and admits a large amount of white space for seemingly little reason.

- The bullet points headers for the paragraphs in the introduction seem distracting to me


line 283: largrangian
line 355: Expertiments
line 385: table table
line 480: Refer 5,6,7,10 in for further results. - Supposed to be reference to some section here?
line 481: AUORC

**Questions:**

See questions posed in the above section.

**Details Of Ethics Concerns:**

-

---

> ### Author Response · Authors · 2024-11-25
>
> We thank the reviewer for taking the time to provide us feedback. We address all the concerns below:
> > The introduction and related work together makes up ~3 pages...
>
> Although these parts are crucial in establishing the relevant context and organization of the paper, we have updated and reduced the content in our related work section in the latest rebuttal revision to make some space for other aspects of the paper such as additional baseline (ANNA in Table 3b).
>
> > It seems rather inefficient to solve the problem for a large number of possible values of $\alpha$....
>
> We would like to clarify that we are NOT training different models for each possible value of $\boldsymbol{\alpha}$. Instead we only add a single classification head for each $\hat{\alpha}\in\boldsymbol{\alpha}$, and the addition in computation time w.r.t. to a single-head model is small, as the added number of parameters is not large w.r.t. the rest of the network. Training different models is a drawback of the method in [1] which we aimed to solve here. We better clarify this in the latest rebuttal revision in section 4.2.
>
> > Experiment details
> >> How many repetitions of experiments were done?
>
> The ‘Wins’ column in Table 2 specifies the exact repetitions done for the experiments across different randomly generated data splits and novel classes (as mentioned in footnote 6 of the manuscript). Further training details are present in appendix selections A.3.1, A.3.2 and in Tables 5-10. For each novel identity, we perform 5 repetitions of the experiment using different data splits generated using random seeds while adhering to the conditions of background shift. We have also updated appendix sections A.3.1 and A.3.2 to explicitly state the above.
>
> >> How is the separability of the novel classes ensured?
> We are unsure whether the reviewer is asking about how separability is ensured in our experiments, or instead in the real world when a user wishes to use our method.
>
> For the first case, as described in section A.3.1, we check the extent to which the novel class can be separated from the other known classes by training a classifier for the task. We add further details to this in the latest rebuttal revision.
>
> In the second case, it is not possible to ensure separability as we do not have labeled data from the novel class. In that case, the separability condition gives us a guarantee on what types of classes we can detect with this method. Or put otherwise, if a class is not detected by the method, then it means it is not separable from the other classes. Anyhow, in our experiments we observe that the assumption is close to being satisfied in some cases like SUN397, and is not perfectly satisfied in scenarios like Amazon Reviews. Having advantageous performance on datasets like Amazon reviews demonstrated the robustness of the method to certain violations of the separability assumption.
>
> > Why does the amount of 'wins' not tally up...
>
> The ‘wins’ do tally up for SUN397 and CIFAR100 dataset. For Amazon Reviews dataset, as mentioned in section 5.4, nnPU and uPU give similar results leading to a tie in their performance for all seeds and novel classes. This is because the objective function of nnPU defaults to that of uPU as the empirical risk remains non-negative.
>
> > Not clear to me how the choice of $\beta$ threshold impacts the results.
>
> Figure 3 in appendix provides further insights on the impact of varying $\beta$ on OSDA performance of CoLOR.
> We have already included the details on selecting FPR based on the threshold $\beta$ in section 4.1 of the submitted manuscript. Moreover,  section 4.1 further states how we can calculate an approximate bound for $\beta$ (constraint on FPR $\hat{\beta}(h)$) using the Rademacher complexity in theorem 1. We find that setting $\beta$ = 0.01 is well within the theoretically calculated bounds and works well in practice for all the experiments. We have included further details about such hyperparameters like $\beta$ in appendix section A.3.2 with Table 4 listing out the exact configurations of all the hyperparameters.
>
> > Unclear what the bolding in table 3b means...
>
> Following the reviewer's comment, we changed this to a more standard bolding strategy since we see the former one can be too complex to parse. Now, the performance of the best performing method is highlighted for each metric and for each setting of distribution shift.
>
> > Figure 2 is small ...
>
> Thank you for the suggestion. We have adjusted the figure sizes in the latest rebuttal revision to make it appropriately large.
>
> > The bullet headers for the paragraphs in the introduction seem distracting...
>
> Thank you for the suggestion. We have updated the paragraph headings in introduction in the latest rebuttal revision to make it less distractive.
>
>
>
> References:
> [1] Yoav Wald and Suchi Saria. Birds of an odd feather: guaranteed out-of-distribution (ood) novel
> category detection. In Uncertainty in Artificial Intelligence, pp. 2179–2191. PMLR, 2023.

---

> > ### Comment · Reviewer_ZrBg · 2024-11-26
> > **Response**
> >
> > I thank the authors for their clarifying response and for making some changes to their paper. I have some further comments and questions.
> >
> > - *Introduction*: I believe that the introduction is still a bit long. While I welcome the removal of the bullet points, I think the headings in the introduction are largely unnecessary. I think the paper would benefit from some further polish in this area.
> >
> > - *Many $\alpha$ values*: Thank you for clarifying this. What would then be a reasonable range of heads that one considers then? You mention a grid search over candidate values $\hat{\alpha}$ in 4.1 and specify the recall target set  $\tilde{\alpha}$ in A.3.2 (I assume that this is the set of candidate values). How would you pick the size of the grid in general and what impact would a coarse grid have? Is there some heuristic one can use?
> >
> > - *Separability in experiments*: I thank the authors for clarifying this. Perhaps I was unclear in my question, I was referring to your experiments. So, when you state in A.3.1 "We then use the learnability
> > of the oracle as a criterion to ensure the separability of novel classes.", what does that actually entail?
> >
> > I also note that I cannot see the authors responses to some other reviewers (Q1zn, skrm and mQDK). Will the authors respond to these too or am I just unable to see these discussions? Will they be made available if that is the case?

---

> ### Author Response · Authors · 2024-11-27
>
> We thank the reviewer for their prompt response to our rebuttal. We have finished the first round of rebuttals for all the reviewers and hope that it is visible to everyone now. We thank the reviewer for their patience and further address the reviewer's concerns below:
>
> > Introduction: I believe that the introduction is still a bit long. While I welcome the removal of the bullet points, I think the headings in the introduction are largely unnecessary. I think the paper would benefit from some further polish in this area.
>
> We have duly noted the suggestions and have reduced the content in introduction and related work sections further to make them concise. We have also removed the headings in the introduction and highlighted the corresponding key insights from the paragraphs instead so that it is easier to follow.
>
> > Many $\alpha$ values: Thank you for clarifying this. What would then be a reasonable range of heads that one considers then? You mention a grid search over candidate values $\hat{\alpha}$ in 4.1 and specify the recall target set $\tilde{\alpha}$ in A.3.2 (I assume that this is the set of candidate values). How would you pick the size of the grid in general and what impact would a coarse grid have? Is there some heuristic one can use?
>
> Intuitively, once the novel class is large enough, then presumably the problem is easier and the difference between CoLOR and baselines like domain discriminator becomes less significant. A domain discriminator, which aims to minimise the loss of classifying between source and target domains, will tend to classify the novel class as “target”, since this class is large and contributes to a large portion of the expected loss in the target domain.
>
> We therefore cap the values of $\hat{\alpha}$ that we iterate through at $0.45$. Regarding grid spacing, we experimented with one dataset to check the sensitivity of results to chosen alpha, and then set the grid for all the experiments based on these sensitivities. We have further added Table 16 in appendix to demonstrate the performance of all the novelty detection heads of CoLOR corresponding to all the candidate target recall constraint values $\hat{\alpha}$ for SUN397 dataset with novel identity images from the set of [water, ice, snow, etc.] and novel class size of $0.06\pm0.01$ for 5 random seeds. Each random seed here corresponds to a unique background shift setting generated randomly by varying the subtype proportions of the known categories (indoor scenes) of SUN397 dataset.
>
> > Separability in experiments: I thank the authors for clarifying this. Perhaps I was unclear in my question, I was referring to your experiments. So, when you state in A.3.1 "We then use the learnability of the oracle as a criterion to ensure the separability of novel classes.", what does that actually entail?
>
> We have further clarified this in our latest revision (section A.3.1).  Particularly, we calculate the AUROC and AUPRC scores of the oracle for the task of supervised novelty detection using the objective of minimising binary cross entropy loss. Higher AUROC and AUPRC would correspond to higher separability. We consider an AUROC and AUPRC higher than 0.98 as ideal to ensure separability between novel class identities and known classes. It is difficult to ensure such high separability for all the datasets, particularly for Amazon Reviews dataset which does not perfectly satisfy the separability. Yet we observe that CoLOR outperforms all the baselines and is robust to background shift in such settings that violate the separability assumption to some extent as observed in Table 2.

---

> > ### Author Response · Authors · 2024-11-30
> > **Following up**
> >
> > Dear Reviewer,
> >
> > We thank you again for your time and comments, we’ve responded to them and made appropriate changes to the paper. We would be happy to see if they alleviate concerns you might have, as we believe they provide an answer to them. With the discussion period deadline approaching soon, we wanted to see if there are any further questions or concerns that you might have, we would be happy to address them up until the deadline.

---

### Official Review · Reviewer_skrm · 2024-11-02

**Soundness:** 2
**Presentation:** 3
**Contribution:** 1
**Rating:** 3
**Confidence:** 3

**Summary:**

This work suggests a new Open-Set Domain Adaptation (OSDA) scenario, “background shift” which is a distribution shift with overlapping supports between source and target distributions exists within the known classes. Also, the existing OSDA methods are experimentally shown to be insufficient in addressing background shifts. To handling this new scenario, this work proposes a new method dubbed as CoLOR. Experimental results show the effectiveness of the proposed method.

**Strengths:**

1. Introducing background shift in OSDA community is interesting and novel.

2. Proposed method is based on theoretical evidence.

3. Experiments are well structured to verify the effectiveness of background shift.

**Weaknesses:**

1. This work has limited novelty and originality since it relies heavily on [1], which provide theoretical analysis of OOD novel category detection. Lemma 1 and Theorem 1 in this manuscript are quite similar claim with Proposition 3.1 and Theorem 4.3 in [1] though they focus different task. Furthermore, it is unclear that Theorem 1 in this manuscript can be easily extended on OSDA since there is no proof in the manuscript and supplementary material.

2. Proposed method is rather incremental. I think it is concatenate the cross-entropy loss on [1].

3. Baselines for performance comparison is insufficient and outdated. I think several recent UDA methods should be included.

[1] Birds of an Odd Feather: Guaranteed Out-of-Distribution (OOD) Novel Category Detection

**Questions:**

1. Why did the authors define background shift not as a distinction between support of $P_{T,[k]}$ and $P_S$ i.e. $Supp(P_{T,[k]}) \neq Supp(P_S)$ but rather as $Supp(P_{T,[k]}) \subseteq Supp(P_S)$????

2. Could you elaborate on the differences in assumptions and results between Theorem 1 in this manuscript and Theorem 4.3 in [1]? If there are differences, please explain how these differences are reflected in the proof process of Theorem 1.

3. I am curious whether the recent UDA methods demonstrate low performance even when applied in OSDA with background shift.

---

> ### Author Response · Authors · 2024-11-27
>
> We thank the reviewer for taking their time and feedback and address all the reviewer's concerns below:
>
> > This work has limited novelty and originality since it relies heavily on [1]...
>
> We note the following:
>
> * Lemma 1 is stronger than Proposition 3.1 in [1]. Both are impossibility statements, but Lemma 1 in our work shows impossibility under an additional assumption of strong positivity. This is crucial, since [2] gives guarantees on OSDA with label shift under this strong positivity assumption, but our lemma shows that this is impossible under the background shifts we consider. Our proof for Lemma 1 can be found in appendix A.1.
> * To extend the guarantees given in Theorem 1 to OSDA, a guarantee on the accuracy of classifying the $k$ known classes is required. This is because Theorem 1 gives a guarantee on classifying the novel class. The guarantee on the remaining $k$ classes can be obtained from classical results in Domain Adaptation, e.g. [3] or others in the survey of Redko et al. [4]. These results show how to relate the expected error of a classifier over a source distribution, to its expected error over a target distribution. We include this detail in our revision, thank you for the comment.
>
> We further highlight the most crucial contributions of our paper below:
> * Existing literature in OSDA overlooks key aspects of distribution shifts where known classes in the target domain have overlapping support with the source domain but still undergo a distribution shift (our work calls this scenario a background shift). Prior work on this is restricted to the label shift setting, however more general background shifts are very relevant in the real world. Our work provides rigorous conditions for successful OSDA under background shift.
> * To the best of our knowledge, existing literature in OSDA under distribution shift within non-novel instances is limited or restricted to specific label shift cases. Our work provides rigorous conditions for separability and background shift, which are widely applicable across real world scenarios where supports of $P_{T,[k]}$ and $P_S$ overlap yet a distribution shift exists.
> * Our work addresses a gap in OSDA studies by exploring the underrepresented scenario of low proportions of unknown objects. We believe this is crucial, as low-proportion settings present unique challenges for OSDA and demand tailored approaches that ensure model reliability without extensive reliance on novel samples.
> * We improve the existing methods with guarantees for novelty detection to scale effectively with large models and for versatile high dimensional datasets like images and text. Such exploration of these learning rules has not been done before, and we think it contributes to the field and we are unaware of prior work that gives finite-sample guarantees for novelty detection in OSDA under distribution shift.
> * Furthermore, as noted by the reviewer, our empirical results rigorously evaluate the joint learning approach, comparing it to OSDA baselines and demonstrating its efficacy under background shift. We believe this analysis is pivotal, as it shows that simply optimizing for both known and novel category detection can outperform traditional, separate approaches to these tasks, an insight we found particularly relevant for OSDA.
>
> We are keen to understand what additional novelty or originality the reviewer deems necessary in light of the contributions outlined here.
>
> > Proposed method is rather incremental ...
>
> The proposed loss function (Eq. 4) in the solution is not a mere concatenation of cross-entropy loss. While cross-entropy loss is used to classify known samples, our approach fundamentally differs from [1] by enabling scalability of the proposed method for high-dimensional data and large models. The learning rule in [1] involves an exhaustive grid search over all possible novel class sizes ($\boldsymbol{\alpha}$), requiring a separate Lagrangian solution for each candidate value $\hat{\alpha} \in \boldsymbol{\alpha}$. Our method addresses this by training a shared representation with multiple novelty detection heads, $h_{\hat{\alpha}}(x_i)$, for each $\hat{\alpha}$ in $\boldsymbol{\alpha}$. This approach amortizes the training time by solving the primal-dual optimization problem only once across all candidate values unlike the learning rule in [1] which requires solving it $\|\boldsymbol{\alpha}\|$ times. This enabled an improved performance even with large data and models, which is further supported by our empirical results. This scalability enhancement represents a core distinction between our method and that of [1].
>
> (Rebuttal continued in next comment...)

---

> ### Author Response · Authors · 2024-11-27
> **Continued...**
>
> > Baselines for performance comparison is insufficient ...
>
> We included ZOC from Esmaeilpour et al. (2022) [6], maximum logit score (MLS) adaptations of ARPL (for SUN397) and ARPL+CS (for CIFAR100) Chen et al. (2021) from Vaze et al. (2022) [7, 5] in Table 2 along with adaptive methods of BODA from Saito et al. (2018) [8], nnPU from Kiryo et al. (2017) [9], uPU from Du Plessis et al. (2014) [10] and DD used in Garg et al. 2021 [12], Du Plessis et al. (2014) [10], Elkan and Noto et al. (2008) [11]. Several of these are quite recent and are popular baselines in the literature. Furthermore, note that we included OSDA baselines like ARPL+CS and ZOC even though they are only applicable to vision tasks, while ours is applicable across modalities. The baselines were chosen to represent methods that are competitive and also represent some methods that are applicable across modalities, like ours. If the reviewer has suggestions for specific additional baselines, we are happy to consider them.
>
> Regarding UDA methods, we are assuming that the reviewer is referring to Unsupervised Domain Adaptation. If the reviewer means Unsupervised Domain Adaptation (UDA) baselines, then please note that it is unclear how one should adapt a UDA baseline to solve an OSDA problem, as these UDA baselines do not handle novel class detection. Therefore UDA baselines are not directly applicable to our problem. Our introduction and related work sections provide an in-depth discussion of recent literature on domain adaptation, OOD detection, novel category detection, PU learning, and OSDA, highlighting the limitations and relevance of these methods to our problem setup. If the reviewer has specific UDA baselines relevant to our problem setup that could provide valuable insights and enhance the quality of our paper, we would be happy to include them.
>
> > Why did the authors define background shif ...
>
> The condition proposed by the reviewer, where $supp(P_{\mathcal{T}, [k]}) \neq supp(P_{\mathcal{S}})$ is much less restrictive than the definition we use in the paper. We defined a shift for which guarantees can be given and is also very reasonable. If we think of novelties as out-of-support examples, then the containment of supports makes sure that $P_{\mathcal{T}, [k]}$ does not introduce any novelties w.r.t $P_{\mathcal{S}}$. In our setting this means that all the novelties are introduced by $P_{\mathcal{T}, [k+1]}$, which is what the definition aims to capture. It is easy to show that if we switch the containment with the condition suggested by the reviewer, then the obtained problem cannot be solved by any algorithm.
>
> > Could you elaborate on the differences in assumptions...
>
> Theorem 1 is a restatement of Theorem 4.3 in [1], specialized to the case of separable novel classes (the theorem in [1] is a bit more general). We use separability for ease of presentation, and since it captures many relevant practical cases. We stated these points in our work (section 3.1), but due to the question raised in the review, we made sure this is further emphasized in our revision.
> Hence the proof process does not differ, and we do not consider theorem 1 as one of our work’s contributions. The novel theoretical component in this work in lemma 1 (proof in appendix section A.1), and our main contributions lie in identification of a practical real world problem of background shift and the proposal of an effective solution that has formal guarantees using existing mathematical tools and is empirically validated using rigorous practical experiment setups which were not explored extensively in the past.
>
> > I am curious whether the recent UDA ...
>
> Similar to our previous response, we again note that it is unclear how one should adapt a UDA (Unsupervised Domain Adaptation) baseline to solve an OSDA problem, as these UDA baselines do not handle novel class detection. Since these are fundamentally different problems, UDA baselines are not directly applicable to our problem. Our introduction and related work sections provide an in-depth discussion of recent literature on domain adaptation, OOD detection, novel category detection, PU learning, and OSDA, highlighting the limitations and relevance of these methods to our problem setup. If the reviewer has specific UDA baselines relevant to our problem setup that could provide valuable insights and enhance the quality of our paper, we would be happy to include them.

---

> > ### Author Response · Authors · 2024-11-27
> > **References**
> >
> > 1] Yoav Wald and Suchi Saria. Birds of an odd feather: guaranteed out-of-distribution (ood) novel
> > category detection. In Uncertainty in Artificial Intelligence, pp. 2179–2191. PMLR, 2023.
> >
> > [2] Saurabh Garg, Sivaraman Balakrishnan, and Zachary C. Lipton. Domain adaptation under open set label shift, 2022. URL https://arxiv.org/abs/2207.13048.
> >
> > [3] Ben-David, Shai, et al. "A theory of learning from different domains." Machine learning 79 (2010): 151-175.
> >
> > [4] Redko, Ievgen, et al. "A survey on domain adaptation theory: learning bounds and theoretical guarantees." arXiv preprint arXiv:2004.11829 (2020).
> >
> > [5] Sagar Vaze, Kai Han, Andrea Vedaldi, and Andrew Zisserman. Open-set recognition: a good
> > closed-set classifier is all you need? In International Conference on Learning Representations,
> > 2022.
> >
> > [6] Sepideh Esmaeilpour, Bing Liu, Eric Robertson, and Lei Shu. Zero-shot out-of-distribution detection based on the pre-trained model clip. Proceedings of the AAAI Conference on Artificial Intelligence, 36(6):6568–6576, Jun. 2022. doi: 10.1609/aaai.v36i6.20610. URL https://ojs.aaai.org/index.php/AAAI/article/view/20610.
> >
> > [7] Guangyao Chen, Peixi Peng, Xiangqian Wang, and Yonghong Tian. Adversarial reciprocal points learning for open set recognition. IEEE Transactions on Pattern Analysis and Machine Intelligence, pp. 1–1, 2021. ISSN 1939-3539. doi: 10.1109/tpami.2021.3106743. URL http://dx.doi.org/10.1109/TPAMI.2021.3106743.
> >
> > [8] Kuniaki Saito, Shohei Yamamoto, Yoshitaka Ushiku, and Tatsuya Harada. Open set domain adaptation by backpropagation, 2018.
> >
> > [9] Ryuichi Kiryo, Gang Niu, Marthinus C Du Plessis, and Masashi Sugiyama. Positive-unlabeled learning with non-negative risk estimator. Advances in neural information processing systems, 30, 2017.
> >
> > [10] Marthinus C du Plessis, Gang Niu, and Masashi Sugiyama. Analysis of learning from positive and unlabeled data. In Z. Ghahramani, M. Welling, C. Cortes, N. Lawrence, and K.Q. Weinberger (eds.), Advances in Neural Information Processing Systems, volume 27. Curran Associates, Inc., 2014.
> >
> > [11] Charles Elkan and Keith Noto. Learning classifiers from only positive and unlabeled data. In
> > Proceedings of the 14th ACM SIGKDD international conference on Knowledge discovery and datamining, pp. 213–220, 2008.
> >
> > [12] Saurabh Garg, Yifan Wu, Alexander J Smola, Sivaraman Balakrishnan, and Zachary Lipton. Mixture proportion estimation and pu learning: A modern approach. Advances in Neural Information Processing Systems, 34:8532–8544, 2021.

---

> > > ### Author Response · Authors · 2024-11-30
> > > **Following up**
> > >
> > > Dear Reviewer,
> > > We thank you again for your suggestions.
> > > With the discussion period deadline approaching, we wish to know if the provided response adequately address your concerns or questions. If there are any further questions or concerns, we would happy to respond up until the deadline.
> > >
> > > Thank you.

---

> > > > ### Comment · Reviewer_skrm · 2024-12-02
> > > >
> > > > I appreciate the authors' efforts in addressing the concerns through their rebuttal. However, I still have several unsolved concerns.
> > > >
> > > > * The authors argue that Lemma 1 is stronger than Proposition 3.1 in [1] because it additionally incorporates the strong positivity assumption. However, I believe that Lemma 1 is actually **weaker** than Proposition 3.1, as it relies on additional assumptions. Since Proposition 3.1 derives conclusions under more general conditions without requiring the strong positivity, it already encompasses the assumptions of Lemma 1. Therefore, Lemma 1 can be inferred from Proposition 3.1, and providing an additional proof for Lemma 1 seems redundant. Moreover, the authors claim that Theorem 1 merely reiterates Theorem 4.3 in [1] and that Lemma 1 represents the novel theoretical component of this study. Given this, I still question the novelty and originality of the work. Can Lemma 1 not be proven as a corollary of the proof for Proposition 3.1 in [1]?
> > > >
> > > > * I also continue to have concerns about the baselines used in the experiments. The OSDA problem targeted in this paper involves: 1) learning a classifier with a labeled source dataset and an unlabeled target dataset, and 2) assigning target instances from known classes to their correct classes while identifying instances from unknown classes as novel classes. **These settings and objectives are shared by both OSDA and UDA**. Hence, I disagree with the author’s statement, "UDA baselines do not handle novel class detection. Therefore, UDA baselines are not directly applicable to our problem." I believe that comparisons with existing methods in both OSDA and UDA are essential. Such comparisons would directly support the authors' claim that "OSDA methods underperform under background shift."
> > > >
> > > > * Furthermore, while the authors claim they used sufficient baselines, **I do not understand why they did not consider methods used as baselines in PULSE [2] such as: UAN [3], DANCE [4], STA [5], and CMU [6]**. I also agree with the direction of developing methods applicable across various modalities. However, this does not mean that comparisons within each modality are unnecessary. If our target task is a vision task and the proposed method performs worse than vision-specific baselines, why should we adopt the proposed approach? Lastly, PULSE [2] conducted experiments not only in vision and text but also in the table modality, which this paper does not address. Thus, I believe this paper does not adequately cover a sufficient range of baselines.
> > > >
> > > > [1] Wald, Yoav, and Suchi Saria. "Birds of an odd feather: guaranteed out-of-distribution (OOD) novel category detection." Uncertainty in Artificial Intelligence. PMLR, 2023.
> > > >
> > > > [2] Garg, Saurabh, Sivaraman Balakrishnan, and Zachary Lipton. "Domain adaptation under open set label shift." Advances in Neural Information Processing Systems 35 (2022): 22531-22546.
> > > >
> > > > [3] You, Kaichao, et al. "Universal domain adaptation." Proceedings of the IEEE/CVF conference on computer vision and pattern recognition. 2019.
> > > >
> > > > [4] Saito, Kuniaki, et al. "Universal domain adaptation through self supervision." Advances in neural information processing systems 33 (2020): 16282-16292.
> > > >
> > > > [5] Liu, Hong, et al. "Separate to adapt: Open set domain adaptation via progressive separation." Proceedings of the IEEE/CVF conference on computer vision and pattern recognition. 2019.
> > > >
> > > > [6] Fu, Bo, et al. "Learning to detect open classes for universal domain adaptation." Computer Vision–ECCV 2020: 16th European Conference, Glasgow, UK, August 23–28, 2020, Proceedings, Part XV 16. Springer International Publishing, 2020.

---

> ### Author Response · Authors · 2024-12-02
>
> We thank the reviewer for their response.
>
> > However, I believe that Lemma 1 is actually weaker than Proposition 3.1, as it relies on additional assumptions. Since Proposition 3.1 derives conclusions under more general conditions without requiring the strong positivity, it already encompasses the assumptions of Lemma 1. Therefore, Lemma 1 can be inferred from Proposition 3.1
>
> This is not true. The impossibility statement of preposition 3.1 in [1] shows that a problem (call it problem A) cannot be solved by any algorithm. This does not mean or imply that the same problem cannot be solved under additional (stronger) assumptions. For example, [2] shows that this problem is solvable under the additional (stronger) assumption of strong positivity when shifts are restricted to label shift. In Lemma 1 of our work, we prove that problem A cannot be solved even when an additional assumption of strong positivity is made. Hence the result we give is stronger and is not a straightforward implication of Proposition 3.1 in [1].
>
> > These settings and objectives are shared by both OSDA and UDA. Hence, I disagree with the author’s statement...
>
> [4,5,6,7] explicitly describe that UDA (Unsupervised Domain Adaptation) follows a strong assumption that the target domain must share the same class space as the source domain while OSDA extends this concept by allowing target domain to contain unknown/novel classes that are not present in the source domain.
>
> Moreover, the reviewer uses an acronym UDA without the full name, which is confusing, since the acronym is most commonly used for a setting without novel classes. Hence, we state that "UDA baselines do not handle novel class detection. Therefore, UDA baselines are not directly applicable to our problem." following from [4,5,6,7].
>
> > However, this does not mean that comparisons within each modality are unnecessary...
>
> Table 3b of our draft includes comparisons (for OSDA under background shift) of recent, popular and best performing OSDA vision-specific baselines like ZOC [9], ARPL from [8], SHOT [7] and ANNA [4] with cross modal methods like DD [13], uPU [12], nnPU [11], CoLOR . Hence, the reviewer's statement that we view comparisons within each modality unnecessary is not correct.
>
> > This paper does not address PULSE [2] ...
>
> As addressed in our paper, PULSE is designed (and provides guarantees) for OSDA under label shift only and does not outperform other methods when there is no label shift [2]. Hence it is reasonable to assume it will not outperform other baselines in the absence of label shift or for shifts that go beyond the assumption of label shift, but we will run it to verify.
>
> > I do not understand why they did not consider methods used as baselines in PULSE [2] such as: UAN (2019), DANCE (2020), STA (2019), and CMU (2020)
>
> We've chosen the best performing baseline from the PULSE [2] paper, along with more modern and popular ones (including but not limited to vision-specific ones) like ZOC [9] (2022), ANNA [4] (2023), SHOT [7] (2021), adaptation of ARPL from [8] (2022) in Table 3b for OSDA under background shift.
>
> (References continued in the next comment)

---

> > ### Author Response · Authors · 2024-12-02
> > **References**
> >
> > [1] Wald, Yoav, and Suchi Saria. "Birds of an odd feather: guaranteed out-of-distribution (OOD) novel category detection." Uncertainty in Artificial Intelligence. PMLR, 2023.
> >
> > [2] Garg, Saurabh, Sivaraman Balakrishnan, and Zachary Lipton. "Domain adaptation under open set label shift." Advances in Neural Information Processing Systems 35 (2022): 22531-22546.
> >
> > [3] Liu, Hong, et al. "Separate to adapt: Open set domain adaptation via progressive separation." Proceedings of the IEEE/CVF conference on computer vision and pattern recognition. 2019.
> >
> > [4] W. Li, J. Liu, B. Han and Y. Yuan, "Adjustment and Alignment for Unbiased Open Set Domain Adaptation," 2023 IEEE/CVF Conference on Computer Vision and Pattern Recognition (CVPR), Vancouver, BC, Canada, 2023, pp. 24110-24119, doi: 10.1109/CVPR52729.2023.02309. keywords: {Adaptation models;Visualization;Computer vision;Codes;Computational modeling;Semantics;Benchmark testing;Transfer;meta;low-shot;continual;or long-tail learning},
> >
> > [5] Saito, K., Yamamoto, S., Ushiku, Y., & Harada, T. (2018).  Open set domain adaptation by Backpropagation. In Proceedings of the European Conference on Computer Vision (ECCV) (pp. 597-613).
> >
> > [6] Luo, Y., Wang, Z., Huang, Z., & Baktashmotlagh, M. (2020). Progressive graph learning for open-set domain adaptation. In International Conference on Machine Learning (pp. 6468-6478). PMLR.
> >
> > [7] Jian Liang, Dapeng Hu, and Jiashi Feng. Do we really need to access the source data? source hypothesis transfer for unsupervised domain adaptation, 2021. URL https://arxiv.org/abs/2002.08546.
> >
> > [8] Sagar Vaze, Kai Han, Andrea Vedaldi, and Andrew Zisserman. Open-set recognition: a good closed-set classifier is all you need? In International Conference on Learning Representations, 2022.
> >
> > [9]Sepideh Esmaeilpour, Bing Liu, Eric Robertson, and Lei Shu. Zero-shot out-of-distribution detection based on the pre-trained model clip. Proceedings of the AAAI Conference on Artificial Intelligence, 36(6):6568–6576, Jun. 2022. doi: 10.1609/aaai.v36i6.20610. URL https://ojs.aaai.org/index.php/AAAI/article/view/20610.
> >
> > [10] Kuniaki Saito, Shohei Yamamoto, Yoshitaka Ushiku, and Tatsuya Harada. Open set domain adaptation by backpropagation, 2018.
> >
> > [11] Ryuichi Kiryo, Gang Niu, Marthinus C Du Plessis, and Masashi Sugiyama. Positive-unlabeled learning with non-negative risk estimator. Advances in neural information processing systems, 30, 2017.
> >
> > [12] Marthinus C du Plessis, Gang Niu, and Masashi Sugiyama. Analysis of learning from positive and unlabeled data. In Z. Ghahramani, M. Welling, C. Cortes, N. Lawrence, and K.Q. Weinberger (eds.), Advances in Neural Information Processing Systems, volume 27. Curran Associates, Inc., 2014.
> >
> > [13] Charles Elkan and Keith Noto. Learning classifiers from only positive and unlabeled data. In Proceedings of the 14th ACM SIGKDD international conference on Knowledge discovery and datamining, pp. 213–220, 2008.

---

### Official Review · Reviewer_mQDK · 2024-11-03

**Soundness:** 2
**Presentation:** 2
**Contribution:** 3
**Rating:** 5
**Confidence:** 3

**Summary:**

This paper addresses Open-Set Domain Adaptation under a specific scenario referred to as "background shift," where known classes have partial overlap between source and target domains, while novel classes remain distinct. The authors introduce CoLOR (Constrained Learning for Open-set Recognition), a multitask learning framework that applies constraints on target data to reduce bias in the feature extractor. The key insight is that traditional partial domain alignment methods struggle when source-private classes significantly outnumber common classes. CoLOR addresses this by jointly optimizing for closed-set classification and novel class detection through a constrained learning approach. The method is evaluated comprehensively across three datasets (CIFAR100, SUN397, Amazon Reviews), demonstrating improved performance, particularly in scenarios with low novel class ratios. The theoretical contributions include a formal analysis of necessary/sufficient conditions for OSDA under background shift.

**Strengths:**

- Strong theoretical foundation with formal analysis of background shift in OSDA, including necessary/sufficient conditions and limitations of existing approaches.
- Comprehensive empirical evaluation across multiple modalities (images and text) and varying novel class ratios.

**Weaknesses:**

- While background shift is justified with potential applications in medical imaging (e.g., identifying known and novel tumor cells), this application is not addressed in the experiments. Including other real-world cases would strengthen the relevance of background shift beyond this initial mention.

- The abstract could be improved by defining “background shift” and avoiding vague phrases like "principled methods."

- The claim “we observe that existing OSDA methods are not robust to the distribution shifts we consider” is strong, but the baselines used are limited. Adding recent OSDA and UniDA baselines would support this claim.
Suggested OSDA baselines: Adjustment and Alignment for OSDA [1], Open-Set Domain Adaptation for Semantic Segmentation [2], Source-Free Progressive Graph Learning for OSDA [3].
Suggested UniDA baselines: Compressive Attention Matching for UniDA [4], Classifiers of Prototypes and Reciprocal Points for UniDA [5].

- The paper assumes “notable separability” between known and novel classes, which might be feasible for images or text using pretrained models like ViT-CLIP. However, this separability may not hold in fields like medical imaging, where pretrained models aren’t available.

- The authors should clarify how they determine the number of classes (k) for any additional classifier head and discuss the impact of varying k.

- In Table 2, BODA outperforms CoLOR on CIFAR100, but this result isn’t highlighted in bold.

- In Table 3, only one baseline is used, which is outdated. The chosen method isn’t clear either, given the three references listed (‘These baselines are domain discriminator (DD), Elkan & Noto (2008); du Plessis et al. (2014); Garg et al. (2021)’).

- Is the “OSCR” metric referring to the commonly used H-score in open-set papers?

- The method is sensitive to hyperparameters like the FPR; details on selecting FPR and setting the threshold  β for optimization should be included.

-  The code is missing .

[1] LI, Wuyang, LIU, Jie, HAN, Bo, et al. Adjustment and alignment for unbiased open set domain adaptation. In : Proceedings of the IEEE/CVF Conference on Computer Vision and Pattern Recognition. 2023. p. 24110-24119.

[2] CHOE, Seun-An, SHIN, Ah-Hyung, PARK, Keon-Hee, et al. Open-Set Domain Adaptation for Semantic Segmentation. In : Proceedings of the IEEE/CVF Conference on Computer Vision and Pattern Recognition. 2024. p. 23943-23953.

[3] LUO, Yadan, WANG, Zijian, CHEN, Zhuoxiao, et al. Source-free progressive graph learning for open-set domain adaptation. IEEE Transactions on Pattern Analysis and Machine Intelligence, 2023, vol. 45, no 9, p. 11240-11255.

[4] ZHU, Didi, LI, Yinchuan, YUAN, Junkun, et al. Universal domain adaptation via compressive attention matching. In : Proceedings of the IEEE/CVF International Conference on Computer Vision. 2023. p. 6974-6985.

[5] HUR, Sungsu, SHIN, Inkyu, PARK, Kwanyong, et al. Learning classifiers of prototypes and reciprocal points for universal domain adaptation. In : Proceedings of the IEEE/CVF Winter Conference on Applications of Computer Vision. 2023. p. 531-540.

**Questions:**

See Weaknesses

---

> ### Author Response · Authors · 2024-11-26
>
> We thank the reviewer for their feedback and time. We would like to address reviewer's concerns below:
>
> > While background shift is justified with potential applications ...
>
> Our setting of OSDA with background shift remains underexplored in research despite it being a common occurrence in practice. As a result there are no benchmark datasets available publicly. An example of its application in medical imaging can be observed in open-set classification of whole-slide tissue images from sentinel lymph nodes of breast cancer patients. Such a setup can be compromised under shifts due to inclusion of either colon lymph nodes data or patient data from other countries or the occurrence of both of these shifts simultaneously [7]. However, such datasets are not publicly available yet.
>
> Moreover, the research in OSDA relies on curation of novel and closed-set classes to simulate shifts and drifts [1,2,3,4,5,6] on real world datasets. This is as such because finding real-world data where the phenomena occur naturally frequently and that these are annotated is very challenging to find. For example, studies [1,4, 5] use Office-Home dataset which consists of images manually curated into four distinct domains (Artistic, Clip Art, Product, and Real-World). This setup does not satisfy Assumption 1 outlined in our manuscript and, therefore, does not represent the background shift scenario we focus on. [3,6] use CIFAR+N and Tiny-ImageNet datasets where novel classes and closed set classes are curated to simulate an OSDA setup.
>
> > The abstract could be improved ...
>
> We appreciate the suggestion. Noted changes have been made in the latest rebuttal revision.
>
> > The claim “we observe that ..." is strong...
>
> We appreciate these suggestions.  We have added these suggested references in the related work section and have included [1] as an additional baseline in our revision. Please see Table 3(b).
> For the rest of the proposed baselines, we restricted our comparison to methods that are applicable across modalities (i.e. both text and image in our experiments). [1, 2, 3] are applied to very specific vision problems, e.g. [2] solves a semantic segmentation problem. Nonetheless we were able to run [1]. At the time of writing this rebuttal [4,5] do not have publicly available code repositories.
>
> > The paper assumes “notable separability” ...
>
> We would be grateful if the reviewer can clarify the question. First, note that separability is a property of the data distribution, not of the model we learned or amount of data we have.
> Hence it is unclear how the availability of pretrained models affect whether or not the assumption holds. Can the reviewer please comment on that?
> If there exists a hypothesis i.e. a function (or a model) that classifies input as known or novel, it is sufficient to say that separability holds.
>
> > The authors should clarify how they determine the number of classes (k) ...
>
> In our experiments, we are assuming that there is a fixed number of known classes (k) and an unknown/novel class. We are not classifying the unknown/novel category into further subcategories.
> In our architecture, we have k known classification heads ($w^c$) and each additional head ($w^\alpha_i$) corresponds to novelty detection constrained under a candidate target recall ($\hat{\alpha}$) from the possible values $\boldsymbol{\alpha}$ such that $\hat{\alpha}\in\boldsymbol{\alpha}$.
>
>
> > In Table 2, BODA outperforms CoLOR ...
>
> Thank you. The change has been made in the latest revision.
>
> > In Table 3, only one baseline is used ...
>
> We included the additional baselines in appendix table 15.
>
> In Table 3 (a) we compare top-1 accuracy over known classes of source-only method with only 2 best performing methods for SUN397 dataset based on the results in Table 2 to optimize for space and include concise relevant information.
>
> Note that Domain discriminator (DD) may be old but still is the second best performing method for SUN397 dataset behind CoLOR. DD is a standard baseline that is adapted and used in all the three references (Elkan & Noto (2008); du Plessis et al. (2014); Garg et al. (2021)) with minor variations. This is similar to how we have adapted it to solve OSDA by including classification heads for known classes along with novel ones. Hence, the three references.
>
> > Is the “OSCR” metric referring to the commonly used H-score ...
>
> These are different metrics.The H-score is computed as the harmonic mean of known class accuracy and unknown class accuracy while OSCR refers to the area under the plot of true classification rate against the false positive rate (misclassifying novel samples as known) [8]. H-score typically requires a fixed threshold for unknown class detection while OSCR is threshold-independent, making it more robust to varying decision boundaries. This property further makes OSCR to be more reliable for evaluating methods under distribution shift settings.
>
> (Rebuttal continued in the next official comment)

---

> > ### Author Response · Authors · 2024-11-26
> > **Continued ...**
> >
> > > The method is sensitive to hyperparameters like the FPR ...
> >
> > We have included the details on selecting FPR based on the threshold $\beta$ in section 4.1 of the submitted manuscript. Moreover, Figure 3 in appendix provide further insights on the impact of varying $\beta$ over the OSDA performance of CoLOR. We have included further details about the hyperparameters in appendix section A.3.2 with Table 4 listing out the exact configurations of all the hyperparameters. We find that setting $\beta$ = 0.01 works well in practice for all the experiments.
> >
> > > The code is missing.
> >
> > We have added the code in the supplementary material (code.zip) and will release it publicly upon the acceptance of the paper.
> >
> >
> > References:
> >
> > [1] Shuo Wen and Maria Brbic. Cross-domain open-world discovery. In Ruslan Salakhutdinov, Zico Kolter, Katherine Heller, Adrian Weller, Nuria Oliver, Jonathan Scarlett, and Felix Berkenkamp (eds.), Proceedings of the 41st International Conference on Machine Learning, volume 235 of Proceedings of Machine Learning Research, pp. 52744–52761. PMLR, 21–27 Jul 2024. URL https://proceedings.mlr.press/v235/wen24b.html.
> >
> > [2] Zongyuan Ge, Sergey Demyanov, Zetao Chen, and Rahil Garnavi. Generative openmax for multi-class open set classification. In Krystian Mikolajczyk and Gabriel Brostow (eds.), British Ma-chine Vision Conference Proceedings 2017, British Machine Vision Conference 2017, BMVC 2017. British Machine Vision Association, 2017. ISBN 190172560X. doi: 10.5244/C.
> > 31.42. URL https://bmvc2017.london/,https://dblp.org/db/conf/bmvc/bmvc2017.html. British Machine Vision Conference 2017, BMVC 2017 ; Conference date:04-09-2017 Through 07-09-2017.
> >
> > [3] Sagar Vaze, Kai Han, Andrea Vedaldi, and Andrew Zisserman. Open-set recognition: a good closed-set classifier is all you need? In International Conference on Learning Representations, 2022.
> >
> > [4] Jian Liang, Dapeng Hu, and Jiashi Feng. Do we really need to access the source data? Source hypothesis transfer for unsupervised domain adaptation, 2021. URL https://arxiv.org/
> > abs/2002.08546.
> >
> > [5] Guangrui Li, Guoliang Kang, Yi Zhu, Yunchao Wei, and Yi Yang. Domain consensus clustering for universal domain adaptation. In 2021 IEEE/CVF Conference on Computer Vision and Pattern Recognition (CVPR), pp. 9752–9761, 2021. doi: 10.1109/CVPR46437.2021.00963.
> >
> > [6] Haoxuan Qu, Xiaofei Hui, Yujun Cai, and Jun Liu. Lmc: Large model collaboration with cross-assessment for training-free open-set object recognition, 2023.
> >
> > [7]  Milda Pocevičiūtė, Yifan Ding, Ruben Bromée, Gabriel Eilertsen, Out-of-distribution detection in digital pathology: Do foundation models bring the end to reconstruction-based approaches?, Computers in Biology and Medicine, 2024, 109327, ISSN 0010-4825, https://doi.org/10.1016/j.compbiomed.2024.109327.
> >
> > [8] Akshay Raj Dhamija, Manuel Günther, and Terrance Boult. Reducing network agnostophobia. Advances in Neural Information Processing Systems, 31, 2018.

---

> > > ### Comment · Reviewer_mQDK · 2024-11-28
> > >
> > > Dear Authors,
> > >
> > > Thank you for addressing my comments and for the effort you’ve put into revising the manuscript. However, there are still points that I find **unclear** or believe **could benefit from further clarification** or correction:
> > >
> > > - The revised abstract is clearer, but there are still inconsistencies. The phrase "**a novel class** along with k non-novel classes" implies only one novel class in the target domain, which seems inconsistent with your experimental setup or OSDA in general. Additionally, **the claim that OSDA methods fail to perform well under covariate shift** feels too strong. Many OSDA methods address covariate shift implicitly through feature alignment or self-training. Could you please clarify or refine this point?
> > >
> > > - Including  **[1], an open-set baseline, in Table 3 (a closed-set scenario) feels misleading**. It would be more appropriate to include it in Table 2. Furthermore, while you restricted comparisons to cross-modal methods, this doesn’t fully justify excluding baselines for vision datasets like SUN397 or CIFAR100. If the suggested baselines aren’t convincing, there are several source-free open-set or universal domain adaptation methods applicable across modalities (with code) that could strengthen your claim about OSDA under covariate shift. I also appreciated your implementation of SHOT (Liang et al., 2021), but showing the results would strengthen the claim about OSDA. Additionally, I don't fully understand your sentence: "but found that the clustering approach performed worse than MSP or entropy-based methods." SHOT is primarily designed for closed-set scenarios, but it has an open-set implementation that uses entropy thresholding to separate known and unknown classes, followed by self-training with clustering for adaptation. Could you clarify what part of SHOT was adapted and why clustering was ineffective?
> > >
> > > - Your assumption that **separability is solely a property of the data distribution**, independent of the model, remains unclear to me. In Table 8, the ZOC method, which performs zero-shot OOD detection leveraging CLIP, outperforms all other methods. For me, this shifts the discussion **from the data distribution’s properties** to the **embedding or feature distribution’s properties**. Could you please provide a simple and clear justification for your separability assumption in this context?
> > >
> > > Overall, I found the **setup and motivation compelling**, but some parts of the work would benefit from stronger justification or clearer evidence. I look forward to your response and feedback from other reviewers before finalizing my rating.

---

> ### Author Response · Authors · 2024-11-29
>
> We thank the reviewer for their active engagement and for providing us the opportunity to address your concerns:
>
> > The phrase "a novel class along with k non-novel classes" seems inconsistent with your experimental setup...
>
> Among the different flavors of open-set problems, the one we study falls in the category where we wish to classify the known classes, and flag examples belonging to the novel ones. It is true that in our setup and others, the novel “class” can consist either of a single category (e.g. Baby, Man, Butterfly etc. in CIFAR100 experiments) or several (say $k_2$) subcategories (e.g. [water, ice, snow, etc.] or [mountains, hills, desert, sky etc.] in SUN397). We emphasize that we are not trying to further separate this novel “class” into its $k_2$ subcategories, which is a different setting some empirical works study. We see how the phrasing in the abstract, “**a novel class along with k non-novel classes**”, may be a bit confusing, therefore we will further clarify by replacing it with “\textbf{a novel/unknown subgroup along with k non-novel classes}” and further clarify this distinction in the main text. Thank you for the comment.
>
> > Many OSDA methods address covariate shift implicitly through feature alignment or self-training.
>
> While many OSDA methods apply measures to deal with some forms of shifts, they usually do not formally define what is the shift they tackle. For instance, one might suspect these existing methods should be robust to label shift, but it turns out they are not [Garg et al.]. Since covariate shift is a wide definition saying that only $p(x)$ changes between source and target, there could be many forms of covariate shift, among them the type of shifts we defined where support overlap holds, and it is unclear whether methods such as feature alignments solve all such covariate shifts. To be more concrete, if we think of feature alignments as finding functions that match the source and target distributions, then consider a simple example where $x$ is a binary variable with source distribution $[0.1, 0.9]$ and target distribution $[0.5, 0.5]$. No function of $x$ can align these distributions, since their supports are equal. Intuitively, methods like feature alignment are most effective when supports do not overlap, and there is some mapping that makes the source and target inputs similar. For instance, in datasets such as Office-Home etc. Due to such subtle implications of giving names to shifts, we find it crucial to formally state what is the case we are solving.
>
> To make the claim in the paper more precise, we will change it to say “OSDA methods underperform under background shift”, which is the setting we formally define and study empirically, where the statement is backed up by our experiments.
>
> > Including  [1] in Table 3 (a closed-set scenario) feels misleading...
>
> We note that Table 3b is indeed an OSDA setting (and not a closed-set setting) for SUN397 dataset as discussed in our draft. While Table 3a does compare the Top-1 accuracies of the closed set classes with additional results in Table 15 to study the impact of background shift on some of the latest methods, Table 3b on the other hand, compares the OSDA performance of various methods with and without background shift (denoted by DS) using evaluation metrics like AUROC (w.r.t. novel classes), AUPRC (w.r.t. novel classes) and OSCR which are only meaningful in the open-set context. We further explicitly mention this in the title of table 3b in the next revision to clarify that the results in Table 3b compare the OSDA performance of the methods. Furthermore, we have indeed explored several latest vision baselines in Table 3b like ZOC [12], adaptation of ARPL from [3], SHOT [13] and ANNA (as recommended by the reviewer) [1] in the OSDA context. In this way we have not restricted ourselves to just cross-modal baselines. We can evidently see that CoLOR mostly outperforms the other baselines for OSDA setting under background shift in Table 3b as well as Table 2.
>
> > showing the results for SHOT [1] would strengthen the claim about OSDA.
>
> We include the results of SHOT baseline below. We will include the results in the final draft (table 3b, table 15) if the reviewer deems it necessary.
>
> **OSDA performance with background shift:**
> | Method         | AUROC  | AUPRC | OSCR | Top-1 Accuracy |
> |----------------|---------------|----------------|---------------|---------------|
> | SHOT  | 0.71 ± 0.16   | 0.19 ± 0.13  | 0.22 ± 0.07  | 0.31 ± 0.05  |
>
> **OSDA performance without background shift:**
> | Method         | AUROC  | AUPRC | OSCR | Top-1 Accuracy |
> |----------------|---------------|----------------|---------------|---------------|
> | SHOT  | 0.61 ± 0.11   | 0.10 ± 0.05  | 0.18 ± 0.06  | 0.30 ± 0.07  |
>
> Note that since SHOT is primarily a source-free domain adaptation method, it is difficult to comment on the impact of distribution shift on its performance on the target domain similar to the case of ZOC [12].

---

> ### Author Response · Authors · 2024-11-30
> **Continued...**
>
> > I don't fully understand your sentence: "but found that the clustering approach performed worse than MSP or entropy-based methods."
>
> To clarify the comment about SHOT baseline, SHOT is a source-free domain adaptation method. However, it does use source data and the source labels to initialize the feature extractor $g_s$ and classifier layer $h_s$ which is called the source model $f_s = g_s \circ h_s$. Then it adapts to the target domain using confidence thresholding strategy by utilizing entropy of model output as uncertainty. We observed that this SHOT strategy to adapt to the target domain (using clustering approach) does not lead to improvement over the OSDA performance of the initial source model using confidence threshold strategy using model output entropies.
>
> > The assumption that **separability is solely a property of the data distribution, independent of the model** remains unclear to me. For me, this shifts the discussion **from the data distribution’s properties to the embedding or feature distribution’s properties**...
>
> We agree that feature representation, model capacity, and dataset size all play a crucial role in our method’s (as well as any others) ability to detect novelties.
> Our only point was to say that mathematically, the separability assumption is a property of the data distribution (considering we control the model class and can choose a flexible neural network). It simply assumes that the model architecture we use is expressive enough to distinguish between the novel class and the rest of the data.  For sure the amount of data will play a crucial role in practice, which is true for any method. But at least under this assumption of separability which says that the novel class can be detected by some hypothetical model, the methods one uses has some finite sample guarantees regardless of the architecture used. It is a common assumption in learning theory (e.g. the realizability assumption in [11]), and seems like a reasonable basic assumption to ground the discussion.
> In practice, things are of course more complex, we must have a large enough sample to learn successfully, though we note that this is true for any other baseline. How well the method performs is then an empirical question, which turns out to have favorable results to CoLOR for many cases.
>
> > In Table 8, the ZOC method, which performs zero-shot OOD detection leveraging CLIP, outperforms all other methods
>
> The high performance of the Zero-shot CLIP (ZOC) method in Table 8 for CIFAR100 is not surprising as CIFAR images are quite simple and have only 1 primary object per image. This is also true for the Tiny-ImageNet (or ImageNet) dataset used in [12] to benchmark ZOC performance. ZOC extends the pre-trained CLIP model by training a text-based image description generator, which aids in predicting unseen classes based on projections from the CLIP image encoder. However, this approach encounters challenges with more complex datasets like SUN397 (Table 3b of our manuscript), which comprises diverse scene categories rather than images centered on a single primary object. The difficulty arises because the text decoder, trained to identify important objects and their relative placements in the image, may not effectively capture the broader contextual semantics inherent in scene images. This limitation suggests that while ZOC excels with datasets featuring clear, singular objects, its performance may decline with images requiring nuanced scene understanding.
>
> **We look forward to your response and engagement from other reviewers as well!**

---

> ### Author Response · Authors · 2024-11-30
> **References**
>
> [1] Shuo Wen and Maria Brbic. Cross-domain open-world discovery. In Ruslan Salakhutdinov, Zico Kolter, Katherine Heller, Adrian Weller, Nuria Oliver, Jonathan Scarlett, and Felix Berkenkamp (eds.), Proceedings of the 41st International Conference on Machine Learning, volume 235 of Proceedings of Machine Learning Research, pp. 52744–52761. PMLR, 21–27 Jul 2024. URL https://proceedings.mlr.press/v235/wen24b.html.
>
> [2] Zongyuan Ge, Sergey Demyanov, Zetao Chen, and Rahil Garnavi. Generative openmax for multi-class open set classification. In Krystian Mikolajczyk and Gabriel Brostow (eds.), British Ma-chine Vision Conference Proceedings 2017, British Machine Vision Conference 2017, BMVC 2017. British Machine Vision Association, 2017. ISBN 190172560X. doi: 10.5244/C. 31.42. URL https://bmvc2017.london/,https://dblp.org/db/conf/bmvc/bmvc2017.html. British Machine Vision Conference 2017, BMVC 2017 ; Conference date:04-09-2017 Through 07-09-2017.
>
> [3] Sagar Vaze, Kai Han, Andrea Vedaldi, and Andrew Zisserman. Open-set recognition: a good closed-set classifier is all you need? In International Conference on Learning Representations, 2022.
>
> [4] Jian Liang, Dapeng Hu, and Jiashi Feng. Do we really need to access the source data? Source hypothesis transfer for unsupervised domain adaptation, 2021. URL https://arxiv.org/abs/2002.08546.
>
> [5] Guangrui Li, Guoliang Kang, Yi Zhu, Yunchao Wei, and Yi Yang. Domain consensus clustering for universal domain adaptation. In 2021 IEEE/CVF Conference on Computer Vision and Pattern Recognition (CVPR), pp. 9752–9761, 2021. doi: 10.1109/CVPR46437.2021.00963.
>
> [6] Haoxuan Qu, Xiaofei Hui, Yujun Cai, and Jun Liu. Lmc: Large model collaboration with cross-assessment for training-free open-set object recognition, 2023.
>
> [7] Milda Pocevičiūtė, Yifan Ding, Ruben Bromée, Gabriel Eilertsen, Out-of-distribution detection in digital pathology: Do foundation models bring the end to reconstruction-based approaches?, Computers in Biology and Medicine, 2024, 109327, ISSN 0010-4825, https://doi.org/10.1016/j.compbiomed.2024.109327.
>
> [8] Akshay Raj Dhamija, Manuel Günther, and Terrance Boult. Reducing network agnostophobia. Advances in Neural Information Processing Systems, 31, 2018.
>
> [9] Yoav Wald and Suchi Saria. Birds of an odd feather: guaranteed out-of-distribution (ood) novel
> category detection. In Uncertainty in Artificial Intelligence, pp. 2179–2191. PMLR, 2023.
>
> [10] Saurabh Garg, Sivaraman Balakrishnan, and Zachary C. Lipton. Domain adaptation under open set label shift, 2022. URL https://arxiv.org/abs/2207.13048.
>
> [11] Shalev-Shwartz S, Ben-David S (2014) Understanding machine learning: from theory to algorithms. Cambridge University Press, Cambridge
>
> [12] Sepideh Esmaeilpour, Bing Liu, Eric Robertson, and Lei Shu. Zero-shot out-of-distribution detection based on the pre-trained model clip. Proceedings of the AAAI Conference on Artificial Intelligence, 36(6):6568–6576, Jun. 2022. doi: 10.1609/aaai.v36i6.20610. URL https://ojs.aaai.org/index.php/AAAI/article/view/20610.
>
> [13] Jian Liang, Dapeng Hu, and Jiashi Feng. Do we really need to access the source data? source hypothesis transfer for unsupervised domain adaptation, 2021. URL https://arxiv.org/
> abs/2002.08546.

---

### Official Review · Reviewer_V3Dn · 2024-11-04

**Soundness:** 2
**Presentation:** 2
**Contribution:** 2
**Rating:** 5
**Confidence:** 3

**Summary:**

This paper addresses a gap in Open-Set Domain Adaptation (OSDA): handling both novel class detection and distribution shifts in known classes ("background shift"). The authors propose a scalable solution combining principled novelty detection with shared representation learning, demonstrating strong results across multiple domains.

**Strengths:**

- identification and formalization of understudied OSDA challenges
- Specific attention to challenging real-world scenarios like low-proportion novel classes

**Weaknesses:**

- All datasets are semi-synthetic. It would be interesting if authors can show results on practical datasets.
- While the problem addressed is quite interesting theoretically, practical relevance is unclear. As mentioned above, the paper may benefit from including results on datasets which inherently satisfy the problem discussed in the paper
- The paper may also benefit from discussing how using stronger CLIP models alters the behavior of the baselines and the proposed method.

**Questions:**

NA

---

> ### Author Response · Authors · 2024-11-24
>
> We thank the reviewer for taking the time to review our work and provide their feedback. Below we have addressed all the concerns:
>
> > All datasets are semi-synthetic. It would be interesting if authors can show results on practical datasets.
>
> Research in OSDA often relies on real-world datasets but where the shifts and drifts are simulated [1, 2, 3, 4, 5, 6]. For example, studies [1,4, 5] use Office-Home dataset which consists of images manually curated into four distinct domains (Artistic, Clip Art, Product, and Real-World). This setup does not satisfy Assumption 1 outlined in our manuscript and, therefore, does not represent the background shift scenario we focus on. [3,6] use CIFAR+N and Tiny-ImageNet datasets where novel classes and closed set classes are curated to simulate an OSDA setup. This is as such because finding real-world data where the phenomena occur naturally frequently and that these are annotated is very challenging to find.
> Our setting of OSDA with distribution shift among the known class samples of source and target data such that their support overlap (background shift), is very natural in practice and remains underexplored in research. As a result there are no benchmark datasets. Some of the naturally occurring examples of background shift scenarios can be observed in open-set classification of whole-slide tissue images from sentinel lymph nodes of breast cancer patients. Such a setup can be compromised under shifts due to inclusion of either colon lymph nodes data or patient data from other countries or the occurrence of both of these shifts simultaneously [7]. However, such datasets are not publicly available yet.
>
> > While the problem addressed is quite interesting theoretically, practical relevance is unclear. As mentioned above, the paper may benefit from including results on datasets which inherently satisfy the problem discussed in the paper
>
> We emphasize again that our setting of OSDA with background shift is a common occurrence in practice and yet, it still remains underexplored in research. Some of the real world examples of such scenarios can be observed in open-set classification of whole-slide tissue images from sentinel lymph nodes of breast cancer patients. Such a setup can be compromised under shifts due to inclusion of either colon lymph nodes data or patient data from other countries or the occurrence of both of these shifts simultaneously [7]. However, such datasets are not publicly available yet.
>
> As mentioned in the previous response, most of the research in OSDA rely on curation of novel and closed-set classes to simulate shifts and drifts [1,2,3,4,5,6]. This is as such because finding real-world data where the phenomena occur naturally frequently and that these are annotated is very challenging to find.
>
> > The paper may also benefit from discussing how using stronger CLIP models alters the behavior of the baselines and the proposed method.
>
> Thank you for the suggestion. For most baselines, pretraining on CLIP and using a ViT-L/14 visual encoder improves performance over ResNet50 (as observed by comparing table 3b and table 2). This performance gain is not surprising due to the richer features obtained from ViT encoder as compared to ResNet50. Following the reviewer’s comments, we’ve further emphasized this in our discussion (appendix section A.2.2).

---

> ### Author Response · Authors · 2024-11-25
> **References:**
>
> [1] Shuo Wen and Maria Brbic. Cross-domain open-world discovery. In Ruslan Salakhutdinov, Zico Kolter, Katherine Heller, Adrian Weller, Nuria Oliver, Jonathan Scarlett, and Felix Berkenkamp (eds.), Proceedings of the 41st International Conference on Machine Learning, volume 235 of Proceedings of Machine Learning Research, pp. 52744–52761. PMLR, 21–27 Jul 2024. URL https://proceedings.mlr.press/v235/wen24b.html.
>
> [2] Zongyuan Ge, Sergey Demyanov, Zetao Chen, and Rahil Garnavi. Generative openmax for multi-class open set classification. In Krystian Mikolajczyk and Gabriel Brostow (eds.), British Ma-chine Vision Conference Proceedings 2017, British Machine Vision Conference 2017, BMVC 2017. British Machine Vision Association, 2017. ISBN 190172560X. doi: 10.5244/C.
> 31.42. URL https://bmvc2017.london/,https://dblp.org/db/conf/bmvc/bmvc2017.html. British Machine Vision Conference 2017, BMVC 2017 ; Conference date:04-09-2017 Through 07-09-2017.
>
> [3] Sagar Vaze, Kai Han, Andrea Vedaldi, and Andrew Zisserman. Open-set recognition: a good closed-set classifier is all you need? In International Conference on Learning Representations, 2022.
>
> [4] Jian Liang, Dapeng Hu, and Jiashi Feng. Do we really need to access the source data? Source hypothesis transfer for unsupervised domain adaptation, 2021. URL https://arxiv.org/
> abs/2002.08546.
>
> [5] Guangrui Li, Guoliang Kang, Yi Zhu, Yunchao Wei, and Yi Yang. Domain consensus clustering for universal domain adaptation. In 2021 IEEE/CVF Conference on Computer Vision and Pattern Recognition (CVPR), pp. 9752–9761, 2021. doi: 10.1109/CVPR46437.2021.00963.
>
> [6] Haoxuan Qu, Xiaofei Hui, Yujun Cai, and Jun Liu. Lmc: Large model collaboration with cross-assessment for training-free open-set object recognition, 2023.
>
> [7]  Milda Pocevičiūtė, Yifan Ding, Ruben Bromée, Gabriel Eilertsen, Out-of-distribution detection in digital pathology: Do foundation models bring the end to reconstruction-based approaches?, Computers in Biology and Medicine, 2024, 109327, ISSN 0010-4825, https://doi.org/10.1016/j.compbiomed.2024.109327.

---

> > ### Author Response · Authors · 2024-11-30
> > **Following up**
> >
> > Dear Reviewer,
> > We thank you again for your time and appreciate your suggestions. With the discussion period deadline approaching, we wish to know if the provided response adequately address your concerns or questions. If there are any further questions or concerns, we would happy to respond up until the deadline.
> >
> > Thank you.

---

### Meta-Review · Area_Chair_XDm6 · 2024-12-22

**Metareview:**

**Summary:** This paper addresses Open-Set Domain Adaptation (OSDA), and proposes CoLOR (Constrained Learning for Open-set Recognition) that integrates novelty detection with shared representation learning. Theoretical guarantees and empirical validation are provided, demonstrating effectiveness on curated datasets under controlled settings.

**Decision:** The paper introduces an interesting and understudied problem, but it falls short on several critical fronts. First, the proposed method and theoretical contributions heavily rely on prior works with limited innovation. The theoretical claims lack rigorous differentiation from prior studies, and some assumptions (e.g., separability) are restrictive and not well-justified. Second, the experiments rely heavily on semi-synthetic datasets without practical real-world benchmarks, undermining the generalizability of the conclusions. Finally, the choice of baselines is insufficient, with several modern OSDA and UDA methods either omitted or inadequately adapted. Besides I noticed that for the concerns about more experimental evaluations, the authors used the lack of publicly available datasets as a justification. However, I personally believe that this precisely highlights that the claim of the paper cannot be fully validated, thereby diminishing its value.

These limitations outweigh the contributions, leading to the decision to reject. During the reviewer-AC discussion period, the reviewers unanimously agreed with this decision.

**Additional Comments On Reviewer Discussion:**

The reviewers engaged actively in the discussion phase, raising concerns about theoretical contributions, baselines, and experimental validation. The authors provided detailed rebuttals and some revisions but failed to alleviate key issues. For example, concerns about the novelty of theoretical contributions, lack of real-world datasets, and limited scalability of the method under large-scale scenarios persisted. While the authors clarified certain points (e.g., separability assumption, hyperparameter sensitivity, semi-synthetic datasets), the responses did not adequately address several key concerns, such as theoretical limitations, evaluations on datasets that satisfy the problem discussed in the paper, further empirical analysis of the robustness of the method when the separability assumption does not hold, etc. As a result, the reviewers unanimously upheld their decision to reject the paper.

---

### Decision · Program_Chairs · 2025-01-22

Reject